# An Ensemble Learning Approach for Reversible Data Hiding in Encrypted Images with Fibonacci Transform

**Shaiju Panchikkil** [1], **Siva Priya Vegesana** [1], **V. M. Manikandan** [1,*], **Praveen Kumar Donta** [2,*], **Praveen Kumar Reddy Maddikunta** [3] and **Thippa Reddy Gadekallu** [3,4]

[1] Department of Computer Science and Engineering, SRM University-AP, Amaravati 522502, India
[2] Distributed Systems Group, TU Wien, 1040 Vienna, Austria
[3] School of Information Technology & Engineering, Vellore Institute of Technology, Vellore 632014, India
[4] Department of Electrical and Computer Engineering, Lebanese American University, Byblos P.O. Box 135053, Lebanon
[*] Correspondence: manikandan.v@srmap.edu.in (V.M.M.); pdonta@dsg.tuwien.ac.at (P.K.D.)

**Abstract:** Reversible data hiding (RDH) is an active area of research in the field of information security. In RDH, a secret can be embedded inside a cover medium. Unlike other data-hiding schemes, RDH becomes important in applications that demand recovery of the cover without any deformation, along with recovery of the hidden secret. In this paper, a new RDH scheme is proposed for performing reversible data hiding in encrypted images using a Fibonacci transform with an ensemble learning method. In the proposed scheme, the data hider encrypts the original image and performs further data hiding. During data hiding, the encrypted image is partitioned into non-overlapping blocks, with each block considered one-by-one. The selected block undergoes a series of Fibonacci transforms during data hiding. The number of Fibonacci transforms required on a selected block is determined by the integer value that the data hider wants to embed. On the receiver side, message extraction and image restoration are performed with the help of the ensemble learning method. The receiver will try to perform all possible Fibonacci transforms and decrypt the blocks. The recovered block is identified with the help of trained machine-learning models. The novelty of the scheme lies in (1) retaining the encrypted pixel intensities unaltered while hiding the data. Almost every RDH scheme described in the literature alters the encrypted pixel intensities to embed the data, which represents a security concern for the encryption algorithm; (2) Introducing an efficient means of recovery through an ensemble model framework. The majority of votes from the different trained models guarantee the correct recovery of the cover image. The proposed scheme enables reduction in the bit error rate during message extraction and contributes to ensuring the suitability of the scheme in areas such as medical image transmission and cloud computing. The results obtained from experiments undertaken show that the proposed RDH scheme was able to attain an improved payload capacity of 0.0625 bits per pixel, outperforming many related RDH schemes with complete reversibility.

**Keywords:** Fibonacci transform; reversible data hiding; data extraction and image recovery; information hiding

## 1. Introduction

In the modern age of technology, most of our day-to-day activities depend on digital applications and technology has significantly influenced the way we perform tasks. With rapid developments in technology, online platforms have become a medium through which we pay our bills, learn and even work. Along with these platforms, cloud technology has become a major resource that we use to store our data, such as pictures, personal information, and professional files [1,2].

The activities we perform online result in the generation of large amounts of data every day. This is not just "any data" but data that has to be kept safe from exploitation [3,4].

Even in online social media networks, people increasingly use images to convey information. When images are stored or transmitted through these public networks, sensitive content may get subjected to attack. Thus, protecting data from attack has become an important task [5].

Data hiding is a technique that helps to protect data by embedding data into a cover medium, such as an image. One of the techniques used is watermarking, where the owner's copyright data is embedded into the image [6]. In case of unauthorized access to the watermarked image, the embedded data is extracted to prove ownership. Embedded watermark data assists in withstanding various forms of attack. A watermarked image remains useless when the system cannot recover the embedded watermark from the processed image, i.e., most details get destroyed. In fragile watermarking, the embedded watermark serves as a verification code, such as an integrity check [7]. When any part of the watermark-protected image shows discrepancies, those parts are subjected to investigation to determine if they have been tampered with. Watermarking techniques may lead to cover image distortion.

Steganography is another widely known data-hiding technique, where the data to be hidden are embedded in a cover medium such that the embedded data are imperceptible to humans. Often the data to be communicated securely are converted into other forms, including encoded or encrypted forms, to make detection of the hidden data detection challenging. The goal of steganography is the secure communication of the secret data and not the cover. This implies that we could adopt any cover to transmit the secret data. The receiver will extract the hidden message from the cover and, during the process, the cover may get distorted, which does not represent a concern from a steganographic point of view [8–10]. This makes reversible data hiding (RDH) different from steganography. Steganography and steganalysis are two sides of the same coin. Steganalysis for hidden message extraction is quite difficult when the data hider embeds less information or when the image is in encrypted form. Machine-learning-based approaches have recently been explored to enable steganalysis to overcome these challenges [11].

RDH schemes are very useful and can be applied in scenarios such as medical image transmission, where the recipient cannot afford any form of distortion in the cover image. In such cases, after extraction of the hidden data the original cover image needs to be restored. Thus, RDH techniques are widely used in medical image transmission. In addition, RDH schemes are useful for forensic and military applications, satellite image transmission, etc., where even slight distortion of the restored image is unacceptable.

RDH can be implemented on a natural image [12–15] or over an encrypted image [16–18]. Drawbacks of using natural images as the cover image include:

1.  The cover image can be easily noticeable and, hence, the contents are exposed to unauthorized users [19].
2.  Embedding of secret data on a cover image modifies the actual pixels of the image, which can pose challenges for reversibility [20].
3.  Such schemes cannot be applied to highly sensitive applications, such as medical image transmission, military operations, satellite image transmission, etc. [21].

Unlike natural images, encrypted images are not informative and, hence, are not easily decodable. Hence, it is much more difficult to identify the presence of any information if the secret information is embedded in an encrypted image. Moreover, encrypting the cover image protects the image from disclosing itself to unauthorized parties, concealing its meaning and, hence, safeguarding its privacy. Thus, the use of RDH in the encrypted domain has become important, attracting researchers to identify new approaches and applications. RDH in the encrypted domain is a challenging task, as the pixel correlation on the cover image no longer exists. Hence, the space for hiding the secret data needs to be vacated, either before encrypting the cover image [22,23] or after the encryption [24,25]. The data hider can embed the secret data into these vacated spaces but RDH requires the receiver to recover the original cover and the hidden secret without any loss. This, in turn, implies that the quantity of data that can be hidden is related to the recovery of the original

cover. For example, the embedding capacity of the RDH scheme described in [26] improves when the cover is processed with a block side-length less than 32, but complete reversibility is guaranteed only at a block side-length of 32 and higher. Thus, a well-designed RDH needs to maintain a positive trade-off between the quantity of the secret and the quality of the recovered cover.

To the best of our knowledge, most of the existing RDH schemes for encrypted images alter the distribution of the pixel intensities of the encrypted image while hiding the secret message [27–29]. Entropy is a metric that measures the randomness of the pixels in an image. The randomness of pixels indicates how different each pixel is from another. Generally, for an encrypted image, the entropy will yield a value near 8 (cover image, being an 8-bit grayscale image) and histogram analysis will generate a flat or uniform histogram. These are indicators of a good encryption algorithm. Conversely, entropy and histogram analysis are good metrics by which to evaluate the quality of the encrypted image. For example, [30] used histogram analysis as one of the metrics to demonstrate that the encrypted image and watermarked image were very different and, hence, that nothing could be inferred about the cover image from the watermarked image. The entropy of the encrypted image and the hidden data image described in [31] are slightly different, demonstrating the efficiency of the scheme. In this context, we propose an RDH scheme for encrypted images which is very different from most such schemes described in the existing literature. The proposed RDH scheme is designed in such a way that the pixel intensities of the encrypted image are never modified during the process of data hiding.

To implement such an RDH scheme, we utilized a well-known image-scrambling algorithm known as the Fibonacci transform. Image-scrambling is treated as a kind of encryption that provides security of the image data by making the contents undecipherable and unreadable. It destroys the high correlation that exists between neighboring pixels. The Fibonacci transform performs scrambling by modifying the pixel locations through simple matrix transformations. Thus, such an algorithm can be tuned for data hiding, meeting our requirement of retaining the pixel intensities of the encrypted image. Many studies are available which address securing data by encryption [32] through the use of pell sequences [33]. Image-scrambling involves matrix transformations that can also be used in information-covering. The Fibonacci transform is an image-scrambling technique that uses a $2 \times 2$ transform matrix with four elements that follows four consecutive numbers from the Fibonacci series [34]. Suppose $(x, y)$ is the original coordinate and a transformation to another coordinate $(x', y')$ is given by,

$$\begin{bmatrix} x' \\ y' \end{bmatrix} = \begin{bmatrix} 1 & 1 \\ 0 & 1 \end{bmatrix} \cdot \begin{bmatrix} x \\ y \end{bmatrix} mod A, \tag{1}$$

where, $A$ is the size of the chosen square matrix. This transformation is called a Fibonacci transformation. We considered the periodicity of the Fibonacci transform to ensure a high payload capacity with full reversibility. Further, in this scheme, an SVM model (support vector machine), a CNN model (convolutional neural network), and a KNN model (k-nearest neighbor) are used to successfully recover the original image and extract the secret information.

### 1.1. Motivation

The present investigation was motivated to address the following key issues:

1. Almost every RDH in encrypted images proposed in the literature modifies the pixel intensities of the encrypted image during the process of data hiding. This is a security concern with respect to the encryption algorithm.
2. The design of an RDH scheme in encrypted images needs to achieve a positive trade-off between the payload and quality of the recovered cover image.

### 1.2. Contribution of the Work

1.  We propose an RDH scheme in encrypted images that is completely reversible with a peak signal-to-noise ratio of $\infty$ and a structural similarity index of 1. The scheme also has an improved payload of 0.0625 bits per pixel.

2.  The use of a Fibonacci transform algorithm for data hiding helps to retain the pixel intensities of the encrypted image during the data-hiding process, thus, preserving the same entropy and histogram of the encrypted image and the image after hiding the data. This helps to support the encryption efficiency and security established via the encryption algorithm.

3.  An efficient framework to achieve complete reversibility is presented. The framework employs ensemble models, whose voting recovers the cover image without any distortion and, simultaneously, the secret message as well.

4.  Our RDH scheme is compatible with any image encryption algorithm and the data recovery depends on the recovery of the cover image. Hence, the embedded secret message is as secure as the cover image.

The rest of this paper is organized as follows: Related studies are discussed in Section 2. The proposed RDH scheme is explained in Section 3. The experimental results are presented in Section 4. The conclusions of the paper, with discussion of future research directions and opportunities, are provided in Section 5.

## 2. Related Work

Many RDH algorithms have been proposed using various approaches, including lossless compression [35], histogram shifting [36–39] and difference expansion [40]. The main idea behind lossless compression is to vacate the room for data embedding. The histogram shifting technique in [36] shifts the zero point and peak point of the histogram to create space to embed the secret data while slightly modifying the pixel values of the original images. Subsequently, a method in which the image is divided into blocks was proposed, where one bit of data is hidden in each of the blocks [26]. This scheme was modified in [27] by introducing a smoothness function that takes border pixels into account, improving the error recovery rate. RDH schemes based on the difference expansion technique [41] embed secret data into the differences between the neighboring pixel values. RDH schemes are also explored for use with different image datasets, such as natural images [42], medical images [43], etc. Though many RDH schemes have been described in the literature [44,45], RDH techniques which operate in an encrypted medium [16,46] have become widely accepted due to their wide range of practical applications pertaining to the security of the cover medium.

To protect the privacy of the cover image, the images are generally encrypted. These encrypted cover images are transferred to applications such as the cloud. This, in turn, protects the image contents from being accessed by an unauthorized party. One of the RDH schemes in encrypted images proposed used a stream cipher to encrypt the image and employed an Arnold transform to hide the secret information [47]. A median edge-detector prediction error-based RDH is discussed in [16]. Prediction errors are encoded using a two's complement encoding approach. A label map of the encoded two's complement is managed for efficient recovery. The label map also needs to be embedded with the additional data as an overhead, but without which the recovery will fail. Another prediction error-based RDH scheme which applies median edge-detector prediction on the adjacent pixels and a quad-tree coding-based room creation for embedding the secret message is described in [17]. Here, the multiple-MSB planes are compressed via quad-tree coding before encrypting the cover image to create the room. In this case, the compression would be maximum on smooth regions and less on rough regions. In [18], a two-tuples coding method is used to compress the MSBs of the cover image to provide adequate room for the additional data during the encryption process. In this scheme, the embedding capacity depends on the rate of compression. A higher embedding rate was achieved with an acceptable recovered quality greater than *PSNR* 40 dB. An RDH scheme in encrypted signals with a public key

cryptosystem was introduced in [48] that was also extended to images. The public key encryption was based on a Paillier homomorphic cryptosystem. The paper was the first to address the issue of allowing the data hider to be anyone. Hence, the security of the scheme depends on the Paillier encryption. Another scheme that addressed the same issue was proposed in [49]. Here, the classical difference expansion was utilized to construct the public key cryptography based on RDH in encrypted images. In [48], the authors describe an approach in which each pixel in the image is divided into two chunks which are encrypted, with the message embedded in each of the pairs. These are added before passing to the receiver. Hence, there is a chance of overflow that was resolved in [49].

In relation to the previous methods, [50] also used a public key cryptosystem, but used histogram-shifting for RDH. The authors used homomorphic multiplication on the encrypted image to expand the histogram and embedded the additional data through histogram-shifting implemented via homomorphic addition. Unlike other schemes, the embedding of different images used a constant of 1bpp based on a histogram expansion that did not rely on the pixel distribution. The authors of [51] combined the lossless and reversible schemes through a public key cryptosystem utilizing the probabilistic and homomorphic properties. In contrast to the other schemes, [52] proposed an improved RDH in encrypted images using a symmetric key cryptosystem. Bits $n$, where $n = 2, 3, 4, \ldots$ are embedded into blocks of fixed side-lengths, by dividing the blocks in terms of $n$ sub-blocks. Hence, the payload was greater and the $PSNR$ was also improved over the block sizes $8 \times 8, 16 \times 16, \ldots$, when compared with [26]. The RDH scheme described in [47] had an overhead of sharing the convolutional neural network model for recovery, which was overcome in [53]. The latter authors utilized the correlation of pixels in the natural images for the recovery of the original cover, as in [26].

An integer wavelet-transform-based RDH in the encrypted domain was proposed in [54]. The orthogonal coefficients after orthogonal decomposition in the high frequency sub-band facilitated the hiding of secret information via a histogram-shifting operation. An adaptive method of data hiding in encrypted images was proposed in [55]. Here, a stream cipher and block permutation was employed to encrypt the cover image. The encrypted blocks were divided into two categories: smooth and complex. A data hider vacated spaces for the additional data by compressing the LSBs of the smooth block, keeping the pixels in the complex block unaltered. In this scheme, the directly decrypted image quality was better as only a few of the portions were used to hide the data.

In the literature on RDH schemes in the encrypted domain, schemes are described that produce a high payload with full reversibility [16–18,23,56]. The intention of the proposed approach here is, however, not to achieve a very high payload but, instead, to achieve a good trade-off between the payload and the quality of the recovered image without changing the pixel values while embedding the data. Such a scheme can safeguard the security established using the encryption algorithm. Moreover, the data embedding is performed over the encrypted image with a Fibonacci transform, which is a widely used image-scrambling algorithm to ensure security.

## 3. Proposed Work

This section discusses the proposed RDH scheme. A schematic diagram of the proposed approach is shown in Figure 1. The first step in the proposed scheme is hiding of the secret information using a Fibonacci transform function. Subsequently, the receiver recovers the original image and hidden secret information using the SVM, CNN, and KNN trained models.

### 3.1. Hiding the Secret Information

In this step, as we are using an RDH scheme on encrypted images, we need to encrypt the original image before embedding the secret information. Let us consider $ORI$ as the original image and the size of the image as $M \times N$. Now generate a pseudo-random matrix

*MM* of the same size $M \times N$ using an encryption key *K1*. To encrypt the image *ORI*, we perform a bitwise XOR operation of *ORI* and *MM*. The resulting encrypted image is *E*.

$$E = ORI \bigoplus MM. \tag{2}$$

The proposed RDH scheme processes the encrypted image *E* by dividing it into different sub-blocks $E_i$, which are of the same fixed size $A \times A$ pixels. Thus,

$$E_i \subset E, \tag{3}$$

and

$$TN = \left\lfloor \frac{(M \times N)}{(A \times A)} \right\rfloor. \tag{4}$$

where, $E_i$ indicates different non-overlapping sub-blocks of *E* of size $A \times A$ pixels. If the total non-overlapping sub-blocks of *E* is *TN* (refer to Equation (4)), then *i* can take a maximum value of *TN*. Hence, we have $0 \leq i \leq TN$. *TN* gives the total number of non-overlapping sub-blocks of size $A \times A$ pixels that we can generate from *E*.

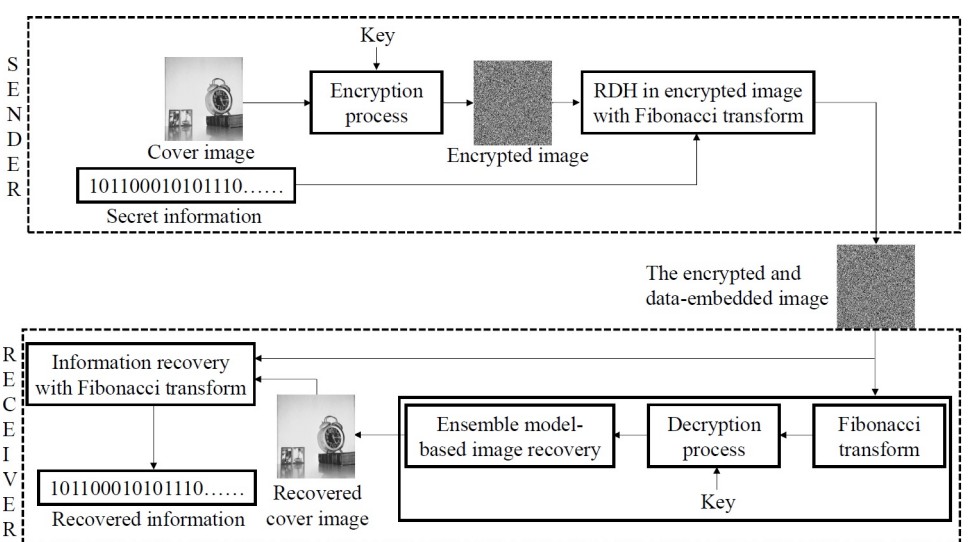

**Figure 1.** Schematic diagram of the proposal.

We considered the standard input image size to the algorithm to be $[512 \times 512]$ for all the images. The encrypted image *E* is divided into *TN* number of non-overlapping blocks with each block size as $A \times A$ pixels.

In addition, we took *A* = 8 for all the images (refer to Section 4.1 to understand the background of fixing the block side-length as 8). The number of bits of data that can be embedded in a block is dependent upon the block size *A*. The Fibonacci transform function has a cyclic period. That is, if the period value is *j* then, after applying the Fibonacci transform *j* times, the shuffled image is again reduced to its original image. So, the values that can be embedded are $\{0, 1, 2 \ldots, j-1\}$. For a block of side length *A*, the period *j* is given by:

$$j = 2 \times A. \tag{5}$$

In the current investigation, the block size considered for the algorithm is 8 and its period value is 16. This implies that we can embed 16 values in the range $\{0, 1, 2 \ldots, 15\}$. Hence, the number of bits *B* that can be embedded per block of size $A \times A$ pixels can be related to the period *j* as:

$$j = 2^B. \tag{6}$$

This implies that we can embed 4 bits of information in a block of side length 8. Consider $I$ as the secret information of length $T$ and each $i^{th}$ individual data value of $I$ is represented as $I_i$. We have,

$$T = TN \times B. \tag{7}$$

Here, $TN$ represents the total non-overlapping blocks of $E$ of side length $A$ (refer to Equation (4)).

Now we pass each $i^{th}$ block $E_i$ and $i^{th}$ individual data value $I_i$ as input to the Fibonacci transform function to perform the scrambling operation on the block using the data unit passed. The Fibonacci transform function generates a scrambled block in return. Hence, the data unit passed is the scrambling factor of the Fibonacci transform function and the number of block scrambles we perform is the basis for embedding the data unit. Similarly, we perform the Fibonacci transform on every block and combine them to form an encrypted and data-embedded image $E'$. The algorithms used to perform the embedding of the secret data in the encrypted image are Algorithms 1 and 2.

### 3.2. Fibonacci Transform Function

The Fibonacci function receives a block $E_i$, a data unit $I_i$ as input, and, depending upon the data unit $I_i$, the block gets scrambled. The image block is taken as a two-dimensional matrix. When the size of the block is $A$, then the block has $A \times A$ elements. Let $x$, $y$ stand for the position of the pixel. Then, $x, y \in \{0, 1, 2 \ldots, A-1\}$. Here for each pair $(x, y)$, after Fibonacci scrambling, becomes $(x', y')$. Hence, the pixels from $(x, y)$ move to the location $(x', y')$ after the scrambling. Similarly, the Fibonacci scrambling function traverses all the pixel points of the block to complete one round of Fibonacci scrambling.

$$\begin{bmatrix} x' \\ y' \end{bmatrix} = \begin{bmatrix} 1 & 1 \\ 0 & 1 \end{bmatrix} \cdot \begin{bmatrix} x \\ y \end{bmatrix} \bmod A$$

Here, if the data unit that has to be embedded is 0, then the Fibonacci transform will scramble the block once. If the data unit is 1, then the block will be scrambled twice. Similarly, if the data needed to be embedded is $n$, then the block will be scrambled $n+1$ times. Thus, the Fibonacci transform function will return a scrambled block.

---

**Algorithm 1:** Securing the secret data.

**Input:** An encrypted image $E$ of size $M \times N$ pixels and the secret information $I$.
**Output:** An image $E'$ which is encrypted and embedded with the secret
       information $I$.
$r = 1$; $s = 1$;
Let $E'$ be an image of size $M \times N$ pixels, initialized with zeros.
Divide the image $E$ into non-overlapping blocks $NB$ of fixed size $A \times A$ pixels.
**while** $(r \leq M - A + 1)$ **do**
    **while** $(s \leq N - A + 1)$ **do**
        Get each block $NB_i$ of size $A \times A$ pixels from $E$. /* $i$ is the index of
           the block $NB$ in $E$.       */
        Get the data unit $I_i$ from the secret information stream $I$. /* $i$ is the
           index of the data processed from $I$.       */
        Call the *FibonacciTransfer* function by passing the block $NB_i$ and the data
           unit $I_i$ as the actual parameters. The /* *FibonacciTransfer* function
           returns a block $NB'_i$ after embedding the data unit $I_i$ in
           $NB_i$.       */
        Place the block $NB'_i$ at the index $i$ of the output image $E'$.
Return the image $E'$ to the receiver.

---

An illustrative example of the data flow while hiding the secret information is shown in Figure 2.

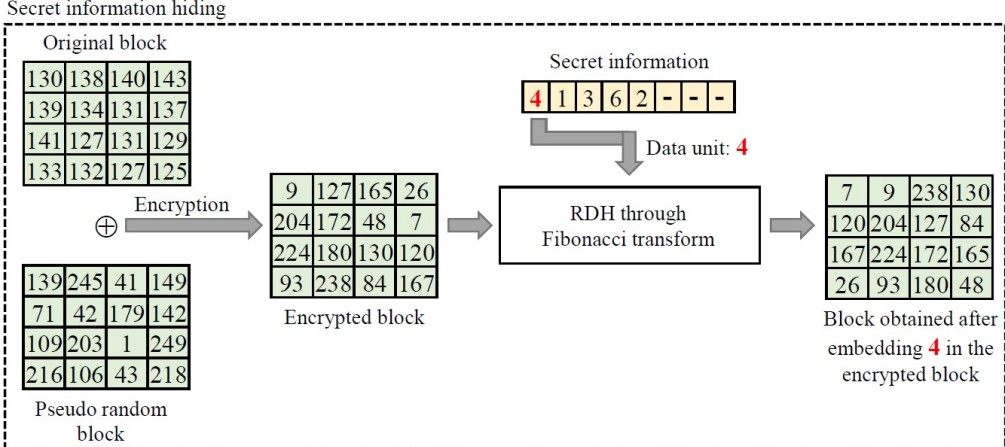

**Figure 2.** Illustration of data flow when hiding the secret information.

---

**Algorithm 2:** FibonacciTransfer function.

---

**Input:** A sub-block $NB$ of size $A \times A$ pixels and the data unit $I$
**Output:** A scrambled block $NB'$
Let $x1 = 1; x2 = 1; x3 = 0; x4 = 1; SS = 0$;
Let $NB'$ be a matrix of size $A \times A$ initialized with zeros;
Let $r = 1; s = 1; k = 1$;
**while** $(SS \leq I)$ **do**
  **while** $(r \leq A)$ **do**
    **while** $(s \leq A)$ **do**
      $xx = mod((x1 \times r + x2 \times s + k \times A), A) + 1$;
      $yy = mod((x3 \times r + x4 \times s + k \times A), A) + 1$;
      $NB'_{xx,yy} = F_{r,s}$;
      $s = s + 1$;
    $r = r + 1$;
  $NB = NB'$;
  $SS = SS + 1$;
Return the scrambled block $NB'$.

---

### 3.3. Recovering the Original Image and Hidden Information

In this phase, the data-embedded image $E'$ and the decryption key $K$ are given as input and we try to output the original image and the hidden secret information. It should be noted that the information shared with the receiver includes the data-embedded image $E'$, the decryption key $K$, and the block size for processing the image $A \times A$. These details are essential to restore the original image and the hidden data.

The receiver begins the recovery process by dividing the image $E'$ into $TN$ non-overlapping blocks of size $A \times A$. The proposed RDH scheme uses machine-learning models, such as support vector machine (SVM), convolutional neural network (CNN), and K-nearest neighbour (KNN) for image recovery. Hence, before we process the non-overlapping blocks of the image $E'$, the machine learning models should be trained as two-class classifiers. Please refer to the details of training the models in Section 3.3.1.

Consider $NB_i$ as the non-overlapping blocks at position $i$ of the image $E'$. This implies,

$$E' = \bigcup_{i=1}^{TN} NB_i. \tag{8}$$

The receiver has to recover each original block from $NB_i$ and also take out the hidden data unit from that block. To achieve this, the receiver processes one block at a time. That is, a block $NB_i$ is passed to the Fibonaccit transform function over a set of possible values

that can be embedded into a block of size $A \times A$ pixels. We know that the data unit that can be embedded in a block of side length $A$ spans in the range $\{0, 1, 2, \ldots, j - 1\}$, where $j$ is the period of the Fibonacci transform (refer to Equation (5)). For each value in the set $\{0, 1, 2, \ldots, j - 1\}$, we have a scrambled form for the block $NB_i$. Let,

$$ZZ = \{0, 1, 2, \ldots, j - 1\}, \tag{9}$$

and the Fibonacci transform function is $F(x, y)$. Thus,

$$F(NB_i, ZZ_l) = NB_{i,ZZ_l}, \tag{10}$$

where $i$ indicates the location of the block $NB$ in $E'$, $ZZ_l$ reflects each individual value from the set $ZZ$ with $0 \leq l \leq j - 1$, and $NB_{i,ZZ_l}$ is the scrambled block generated by passing $NB_i$ and $ZZ_l$ as the arguments to the Fibonacci transform function.

For example, if we assume that the block size $A = 8$, then the maximum values that the Fibonacci transform can embed in these blocks is from $\{0, 1, 2, \ldots, 15\}$. Hence, we perform the Fibonacci transform function on each block over this set of data and, thus, return 16 different versions of the same block.

The next step in the process is to decrypt all the different $NB_{i,l}$ blocks and to identify the original block. Let the decrypted form of the $NB_{i,l}$ be indicated as $DB_{i,l}$. The function of the ensemble approach is to identify the original block from the different $DB_{i,l}$ blocks. The ensemble algorithm works in such a way that a block is taken into confidence if at least two of the models favor or classify a particular block $DB_{i,l}$ as the original block. In practice, the SVM and KNN models take the feature vectors of the blocks to perform classification, whereas the CNN model takes in the blocks directly to extract their own features and classify the blocks.

Here, the RDH scheme considers a voting point of view to find the original block. It considers a block to be original if it is classified as an original block by the majority of the working models. That is, the block is classified as original if all the three or at least two models classify it as an original block. In the worst-case scenario, the block which is classified as an encrypted block with the least confidence by the SVM and CNN models is taken into confidence. An illustrative example of the original block-recovery process at the receiver side is shown in Figure 3.

Now, let the recovered original block be $RB_i$. Thus,

$$RB_i \in DB_{i,l}. \tag{11}$$

Once the original block $RB_i$ is recovered, the hidden information in the $NB_i$ of the image $E'$ can be taken out. We follow the steps below to take out the hidden information in each of the $NB_i$ blocks of $E'$.

1.  Encrypt the recovered block $RB_i$ using the encryption key. This generates an encrypted block, say $EB_i$.
2.  Pass $EB_i$ and each data unit from the set $ZZ = \{0, 1, 2, \ldots, j - 1\}$ as arguments to the Fibonacci transfer function, where $j$ is the period of the Fibonacci transfer function. Referring to Equation (10), we have

$$F(EB_i, ZZ_l) = NEW_{i,l}, \tag{12}$$

3.  Check for the equality between the scrambled block $NEW_{i,l}$ and $NB_i$.
4.  We stop the process after finding a match. The corresponding value $l$ indicates the data hidden in that particular block $NB_i$.

We continue the process by recovering all the original $TN$ blocks along with the recovery of hidden data units in each block to generate the original cover image and the stream of secret information. An illustrative example of the secret information recovery

process is shown in Figure 4. The detailed process of image recovery and secret data extraction is also given in Algorithm 3.

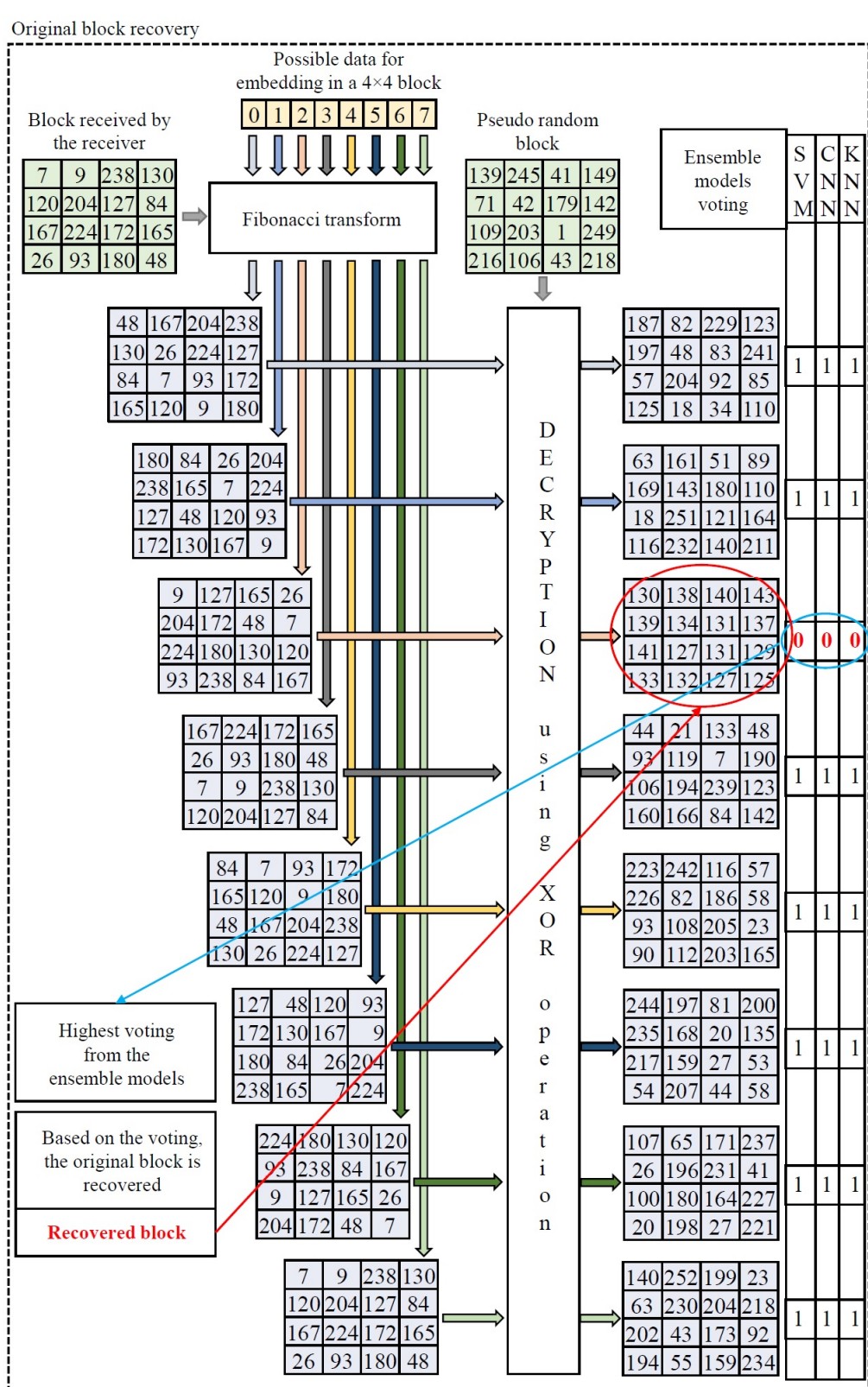

**Figure 3.** Illustration of original block-recovery process.

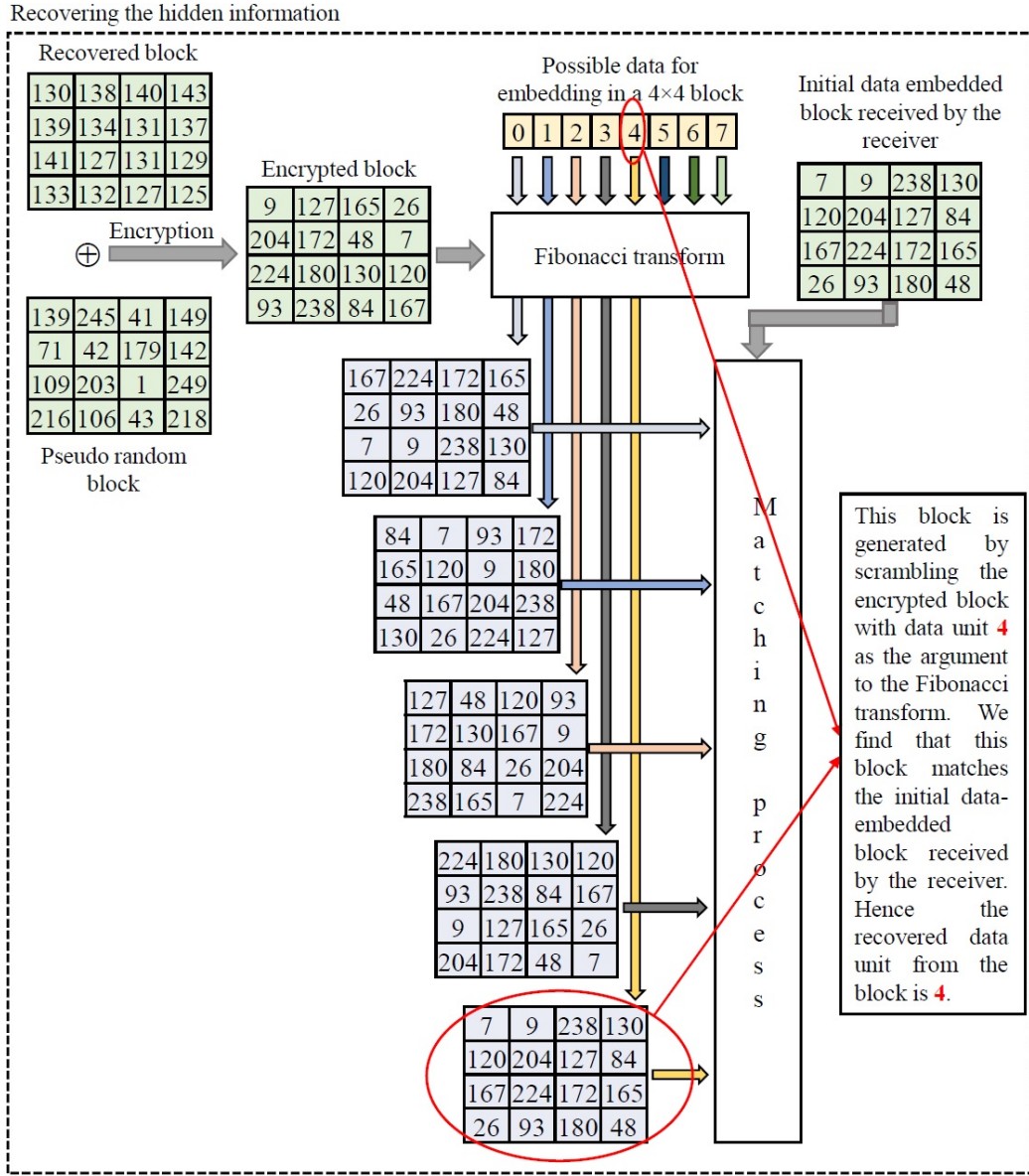

**Figure 4.** Illustration of the recovery of the hidden secret information.

### 3.3.1. Training of the Machine-Learning Models

The feature set to train and validate the different ensemble learning algorithms is selected from four different images that are randomly chosen from the USC-SIPI image database. The SVM model and the KNN model are trained on a set of five feature vectors. For this, we computed the feature vector on 240 different blocks of size $8 \times 8$ pixels. From the selected set of 240 blocks, 120 are original and the rest of the 120 are encrypted blocks. The third model, called CNN, is trained to act as a binary classifier by taking the original and encrypted blocks directly as input. The CNN model is trained with $16,384$ blocks. A total of 75% of these blocks were used for training the model and the rest of the 25% blocks were used for validation.

---

**Algorithm 3:** Ensemble based recovery.

---

**Input:** An image $E'$ of size $M \times N$ pixels which are encrypted and embedded with the secret information $I$, feature vector table, classification table, a folder with the original and encrypted blocks, and the decryption key $K$.

**Output:** The recovered original image $IM$ and the recovered information stream $I$.

Let $IM$ be the recovered image of size $M \times N$ initialized to zero.

Let $SVMModel$ be the trained support vector machine model on input feature vectors and class.

Let $KNNModel$ be the trained K-nearest neighbor model on input feature vectors and class. Set the number of neighbors $k = 10$.

Let $CNNModel$ be the trained convolutional neural network model on the original and encrypted blocks.

Divide the image $E'$ into non-overlapping blocks $NB$ of fixed size $A \times A$ pixels.

Generate a pseudo-random matrix $G$ of size $M \times N$ pixels whose values are within [0–255] using the decryption key $K$.

$r = 1; s = 1;$

**while** $(r \leq M - A + 1)$ **do**

    **while** $(s \leq N - A + 1)$ **do**

        Get each block $NB_i$ of size $A \times A$ pixels from $E'$. /* $i$ is the index of the block $NB$ in $E'$. */

        Get each block $G_i$ of size $A \times A$ pixels from the pseudo-random matrix $G$. /* $i$ is the index of that block in $G$. */

        $t = 1; RB = B_i$

        **while** $(t \leq A)$ **do**

            Pass the block $NB_i$ and $t$ to the $FibonacciTransform$ function. The /* $FibonacciTransform$ function returns a block $NB_i^t$ after embedding $t$ in the block $NB_i$. */

            Decrypt the block $NB_i^t$ by XORing with $G_i$. /* Let the decrypted block be $DB_i^t$. */

            Compute the feature vector $F$ of $DB_i^t$.

            Predict the $label1$ and $confidence1$ by passing $F$ to $SVMModel$.

            Predict the $label2$ and $confidence2$ by passing the block $BB_i^t$ to $CNNModel$.

            Predict the $label3$ and $confidence3$ by passing $F$ to $KNNModel$.

            /* The confidence measure is a vector with two different values that correspond to the highest confidence value and lowest confidence value. */

            If $label1$, $label2$, and $label3$ gives positive votes for the block $B_i^t$ then the recovered block $RB = DB_i^t$.

            If the above condition fails, we check if any of the two labels give positive votes. If so, we preserve that recovered block as $RB = DB_i^t$.

            In the worst case, when the above two conditions do not hold, we find the lowest confidence measure of each block via SVM prediction $RB1$ and CNN prediction $RB2$. We preserve the block which has the lowest confidence among the two predictions $RB = RB1$ or $RB = RB2$. [Here, the lowest confidence implies the lowest measure with which the model classifies a block as an encrypted block]

            $t = t + 1;$

        Preserve the recovered block $RB$ at the index $i$ of the image $IM$.

        Each embedded data value $I_i$ is extracted by passing the encrypted form of the recovered block $RB$ to the Fibonacci transform and comparing the resulting block with the corresponding block of $E'$.

        $r = r + 1;$

    $s = s + 1;$

---

Below are the five feature vectors that are considered to train the SVM and the KNN models. The training is performed to make the model distinguish each block into original and encrypted blocks. They are:

1. F1 (Entropy): The entropy of the image is defined as the degree of randomness. The original image has lesser entropy than the encrypted image. Thus, entropy acts as an important feature for the classification of the blocks.
2. F2 (Standard Deviation): The standard deviation measures the deviation of the measured values or the data from its mean. Here, the standard deviation measures the deviation of pixels from their mean. The higher the value, the more encrypted the block.
3. F3 (Smoothness): The smoothness measure helps to understand the correlation between the adjacent pixels of the image. Hence, the encrypted blocks have a high smoothness value when compared to the original blocks.
4. F4 (Difference in Gray-Level): The difference between the maximum gray level and the minimum gray level obtained from the histogram is known as the difference in gray level.
5. F5 (Average Gradient): The average gradient considers the variation in each of the adjacent pixels. Hence, it can be a useful feature for the classification of blocks.

## 4. Experimental Results

The operation of the proposed reversible data-hiding scheme was analyzed by testing its results on the $USC - SIPI$ dataset [57]. The $USC - SIPI$ dataset is a digitized image data set provided by the University of Southern California which contains 210 images. Of these 210 images, 38 are aerials, 69 are sequences, 64 are textures and 39 are miscellaneous images. Here, during the testing, we considered the block size for all the images as 8, i.e., $A = 8$, and all the results are for the same block size. The experimental study was performed on a computer with processor Intel(R) Core(TM) i5-9300H CPU @ 2.40 GHz, and 8.00 GB installed RAM. The operating system was a 64-bit operating system. The simulations were carried out using the MATLAB software tool, version R2019a.

### 4.1. Computing the Image-Processing Block Size

Section 3.2 gives an outline of the Fibonacci transfer function, which is the key concept used for embedding the secret data in the proposed approach. The maximum information that we can embed in a particular block of size $A \times A$ pixels depends purely on the periodicity of the transfer function. The periodicity of the Fibonacci transfer function depends on the transformation matrix used in the algorithm. In this investigation, the transformation matrix used in the transfer function is $\begin{bmatrix} 1 & 1 \\ 0 & 1 \end{bmatrix}$ (refer to Section 3.2). The embedding capacities that can be achieved with various block sizes are given in Table 1. It can be seen from Table 1 that the payload rate decreases as the processing block size of the image is increased. The recovery factor is taken into account, along with the payload rate, to fix the processing block size of the images. The image recovery rate is explained in Section 4.7. Based on consideration of the factors referred to, we chose $8 \times 8$ as the feasible block size for processing the images.

**Table 1.** Relation between the periodicity of the Fibonacci transform and the block size.

| Sl. No. | Block Size | Periodicity | Bits Embedded in a Block | Payload Achieved (bpp) |
|:---:|:---:|:---:|:---:|:---:|
| 1 | $8 \times 8$ | 16 | 4 | 0.0625 |
| 2 | $16 \times 16$ | 32 | 5 | 0.0195 |
| 3 | $32 \times 32$ | 64 | 6 | 0.0059 |

## 4.2. Embedding Capacity

The embedding capacity represents the number of bits of information that can be embedded in a pixel of the image. The embedding capacity is calculated as the total number of bits embedded in the image divided by the total number of bits that represent the image. Thus, the units are described as bits per pixel (bpp).

While using this RDH Scheme, we embedded four bits of data in each $8 \times 8$ block. So, the embedding capacity for the scheme was 0.0625 bpp.

## 4.3. Error Rate

The error rate is the measure that indicates the number of incorrectly recovered bits in the extracted secret information. Here, the RDH scheme is able to recover with a zero-bit error rate, which implies that it extracts the embedded data error-free.

The details of the embedding capacity and the bit error rate of the proposed RDH scheme are represented in Table 2.

**Table 2.** Rate of information embedded and bit errors on airplane, baboon, boat and peppers images.

| Name of the Image | Block Size | Data Embedding Rate (bpp) | BER |
|---|---|---|---|
| Airplane | $8 \times 8$ | 0.0625 | 0 |
| Baboon | $8 \times 8$ | 0.0625 | 0 |
| Boat | $8 \times 8$ | 0.0625 | 0 |
| Peppers | $8 \times 8$ | 0.0625 | 0 |

## 4.4. Restored Cover Image Analysis

It is important to evaluate the quality of the restored image when compared with the original cover image. The reversibility cannot be guaranteed if the restored image is not the same as the original cover image. The quality of the restored image is determined using the $PSNR$ and the $SSIM$ values. The $PSNR$ is the peak signal-to-noise ratio, which is a measure of the ratio of the maximum possible gray level of the image to the noise distortion that degrades the image quality. The units are decibels (*dB*). For a fully recovered image, if the mean squared error is zero, then the expected ideal value for the $PSNR$ is $\infty$ [58,59].

$SSIM$ is the structural similarity index. It is a measure of the quality of the restored image based on visual perception. The $SSIM$ value is 1 when the original and the recovered images are structurally similar.

The $PSNR$ and $SSIM$ results for the four images, the baboon, the airplane, the boat, and the peppers, are given in Table 3. Some output results for the images that were generated during the experiment are shown in Figure 5.

**Table 3.** $PSNR$ and $SSIM$ on airplane, baboon, boat and peppers images.

| Name of the Image | Peak Signal-to-Noise Ratio (*PSNR*) | SSIM |
|---|---|---|
| Airplane | $\infty$ | 1 |
| Baboon | $\infty$ | 1 |
| Boat | $\infty$ | 1 |
| Peppers | $\infty$ | 1 |

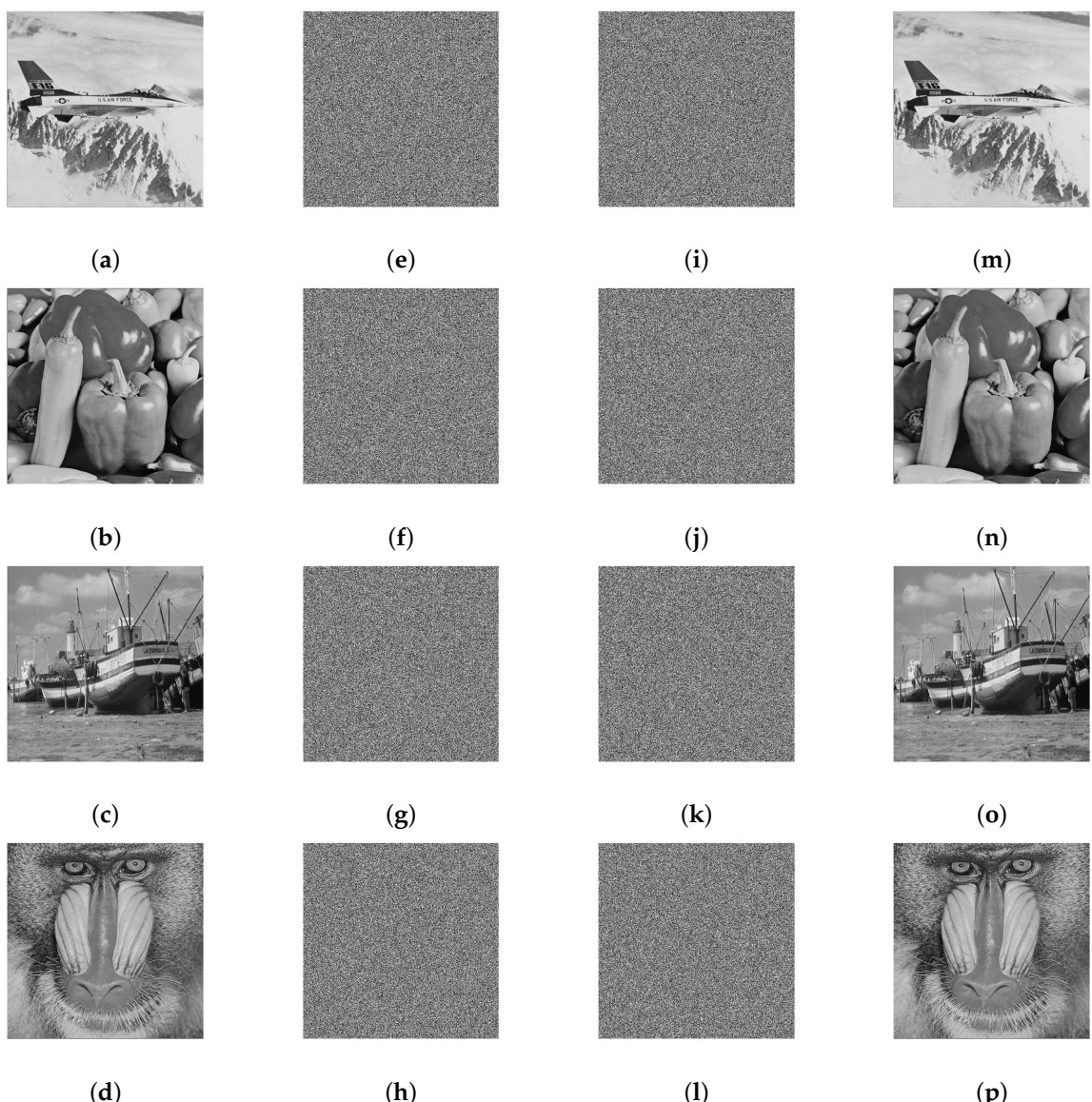

**Figure 5.** Intermediate outputs generated during the experiments: Figure (**a–d**): Original images, Figure (**e–h**): Encrypted images, Figure (**i–l**): Images after the embedding process, and Figure (**m–p**): Recovered images.

### 4.5. Entropy of Images

An image of high randomness is desirable for communication as the pixels in it are indistinguishable in nature. The purpose of using an encryption algorithm is to generate an encrypted image from which no informative patterns about the pixels can be inferred [60]. Hence, the benefit of using an encryption algorithm is the production of an image with maximum pixel randomness [61]. This can be measured using a quantitative measuring metric denoted entropy. Entropy is a statistical measure of randomness that can be used to characterize the texture of the input image [62]. Here, as the pixel values extend up to 256, the maximum entropy values should be near 8. The entropy of the encrypted images is greater than the entropy of the original images. The entropy of an image $E_I$ can be calculated as:

$$E_I = \sum_{n=0}^{Pk-1} P(n) \log_2 \frac{1}{P(n)},$$  (13)

where $Pk$ denotes the peak intensity level and $P(n)$ gives the probability of the occurrence of the pixel $n$. The entropy values of the original image, encrypted image, and the information-

embedded image for the images of baboon, airplane, boat, and peppers are presented in Table 4.

**Table 4.** Entropy measured on airplane, baboon, boat and peppers images.

| Name of the Image | Entropy (Original Image) | Entropy (Encrypted Image) | Entropy (Data Embedded Image) |
|---|---|---|---|
| **Airplane** | 6.7025 | 7.9992 | 7.9992 |
| **Baboon** | 7.3583 | 7.9994 | 7.9994 |
| **Boat** | 7.1914 | 7.9993 | 7.9993 |
| **Peppers** | 7.5937 | 7.9992 | 7.9992 |

### 4.6. Histogram of Images

The histogram analysis helps to analyze the uniform distribution of the pixel intensities, which are important to conceal the image from attacks. The histograms for the encrypted image and the information embedded image are flat, which helps in protecting images from attack. The histograms for the *Airplane*, *Baboon*, *Boat* and *Peppers* images are shown in Figure 6. The figure shows the histogram outputs for the original, encrypted, and data-embedded images.

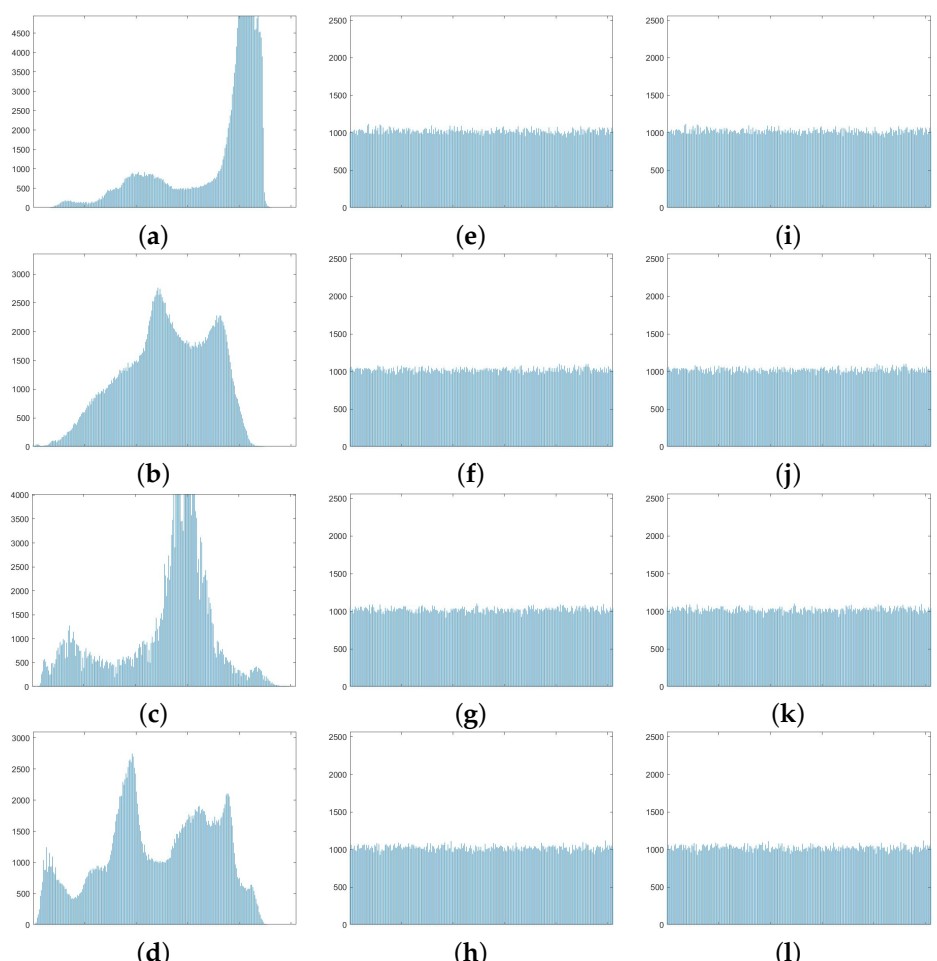

**Figure 6.** Histogram results of the four images: airplane, baboon, boat and peppers. Figure (**a**–**d**): Original image histograms, Figure (**e**–**h**): Encrypted image histograms, Figure (**i**–**l**): Histograms from the images after hiding the data.

### 4.7. Recovery of Images

The recovery of images analyzed on 210 images from the USC-SIPI dataset is described here. The proposed RDH algorithm was able to recover 97.37% of aerials, 98.55% of sequences, 67.19% of textures, and 94.87% of miscellaneous images, successfully. The overall recovery for all the 210 image dataset was 88.1%. Figure 7 depicts the results related to the image recovery according to the image category.

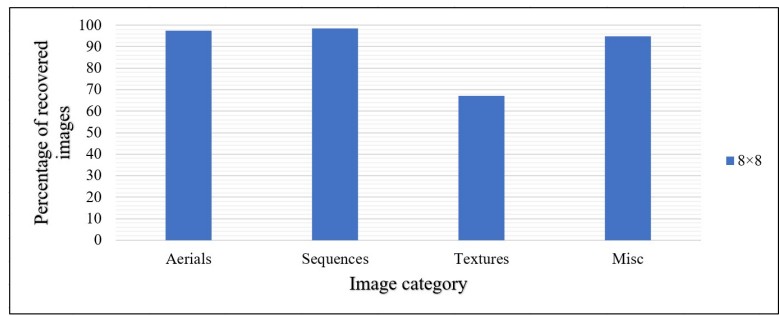

**Figure 7.** Rate of images recovered.

### 4.8. Comparative Study

We compared the embedding capacity of our scheme with that of similar RDH schemes. The comparison chart is shown in Table 5. As observed, we were able to achieve an embedding capacity of 0.0625 bpp, which was better than that of several existing schemes. Apart from the embedding rate, all the compared schemes recovered the original cover image with a PSNR of $\infty$ and $SSIM = 1$.

It should be noted that there are other RDH schemes in encrypted images [16–18,23,56], that achieve a very high payload, but the proposed scheme is very different from most of the earlier schemes. The main objective of retaining the pixel distribution of the encrypted image while embedding the data was met. An acceptable payload of 0.0625 bpp was achieved without compromising the complete reversibility feature.

**Table 5.** Comparison of payload computed on airplane, baboon, boat and peppers images with similar RDH schemes.

| Scheme | Airplane | Baboon | Boat | Peppers |
|---|---|---|---|---|
| Scheme in [26] | 0.0039 | 0.0009 | 0.0039 | 0.0015 |
| Scheme in [27] | 0.0039 | 0.0009 | 0.0039 | 0.0039 |
| Scheme in [29] | 0.0300 | 0.0100 | 0.0300 | 0.0300 |
| Scheme in [48] | 0.0020 | 0.0020 | 0.0020 | 0.0020 |
| Scheme in [49] | 0.0040 | 0.0040 | 0.0040 | 0.0040 |
| Scheme in [51] | 0.0080 | 0.0080 | 0.0080 | 0.0080 |
| Scheme in [50] | 0.0080 | 0.0080 | 0.0080 | 0.0080 |
| Scheme in [30] | 0.0039 | 0.0039 | 0.0039 | 0.0039 |
| Scheme in [28] | 0.0312 | 0.0312 | 0.0312 | 0.0312 |
| Scheme in [54] | 0.0400 | 0.0400 | 0.0400 | 0.0400 |
| Scheme in [47] | 0.0547 | 0.0547 | 0.0547 | 0.0547 |
| Scheme in [55] | 0.0240 | 0.0045 | - | - |
| Scheme in [52] | 0.0047 | 0.0011 | 0.0072 | 0.0072 |
| Scheme in [53] | 0.0547 | 0.0547 | 0.0547 | 0.0547 |
| Proposed scheme | 0.0625 | 0.0625 | 0.0625 | 0.0625 |

## 5. Conclusions

We proposed a novel reversible data-hiding scheme in encrypted images using the Fibonacci transform function. The entropy values and histograms of the encrypted and information-embedded images are retained through the Fibonacci transform-based data embedding process, which ensures image security. We used the machine learning models, support vector machine, convolutional neural network, and K-nearest neighbor, to recover the original image without any distortion. The proposed RDH scheme uses the Fibonacci transform function with an image-processed block size of $8 \times 8$ pixels, ensuring an embedding capacity of 0.0625 bpp, which is better than that of many similar state-of-the-art algorithms. The scheme is able to ensure a lossless recovery of a majority of the images from the 210 different images of the USC-SIPI image database with a bit error rate of 0, a peak signal-to-noise ratio of $\infty$, and a structural similarity index of 1. The scheme can also be tested on other cover mediums, such as video, in future work.

**Author Contributions:** Conceptualization, S.P. and S.P.V.; Data curation, V.M.M.; Formal analysis, P.K.R.M., S.P. and S.P.V.; Investigation P.K.D.; Methodology, P.K.D., S.P. and S.P.V.; Resources, P.K.D.; Software, P.K.D.; Supervision, V.M.M.; Visualization, P.K.R.M; Writing—original draft, V.M.M., P.K.R.M. and T.R.G.; Writing—review & editing, V.M.M. and P.K.D. All authors have read and agreed to the published version of the manuscript.

**Funding:** This research received no external funding.

**Data Availability Statement:** Not applicable.

**Conflicts of Interest:** The authors declare no conflict of interest.

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
