# Peer review of "An Ensemble Learning Approach for Reversible Data Hiding in Encrypted Images with Fibonacci Transform"

_electronics, doi:10.3390/electronics12020450_

Round 1

Reviewer 1 Report

Please find the attached report. 

Author Response

Review (i) In this paper, authors have used Fibonacci transform function for data hiding but I could not find the definition of Fibonacci transform function in the paper. It is my suggestion that authors should at least provide this definition and some relevant reference(s) where Fibonacci or other famous sequences such generalised Fibonacci or Pell sequence has application in information security and other areas.

Authors may consult the following references for Fibonacci and Pell sequences

  1. Text Encryption Using Pell Sequence and Elliptic Curves with Provable Security
  2. Horadam generalized Fibonacci numbers and the modular group
  3. Pell numbers, Pell–Lucas numbers and modular group

Reply (i): We thank the reviewer for highlighting the importance of giving a note on the image scrambling and Fibonacci transform in the Introduction Section. Accordingly, we have updated the contents of the Introduction Section, by including the definition for Fibonacci transform as well. We have also cited the mentioned articles at line numbers: 94-101 of the updated manuscript.

Review (ii) It would be better to write the formula for the Entropy in Section 3.4.

Reply (ii): We have included the formula for the Entropy in Section 3.4. The Section 3.4 is changed to Section 3.5 in the revised manuscript.

Review (iii) Why only the histogram of the Baboon is given in Fig 6? It would be better to

present histograms of the airplane, boat and pepper as well.

Reply (iii): We thank the reviewer for asking to include the histograms of the Airplane, Baboon, Boat and Peppers images. As per the suggestion, we have updated Fig. 6. 

Review (iv) Explain PSNR and the SSIM tests.

Reply (iv): Peak signal to noise ratio (PSNR) and structural similarity index (SSIM) are the two well-known image quality measures to compare a possibly modified image with the original image. The researchers working on data hiding schemes are widely using this metric to evaluate the visual quality of the stego image by comparing it with the original cover image. In reversible data hiding context, the quality of the restored image can be evaluated by comparing it with the original image using PSNR and SSIM measures. If the restored image is exactly same as the original image then we shall observe a PSNR of ∞ and SSIM of 1. A brief description of PSNR and SSIM are also given in the manuscript for better understanding. 

Review (v) There is a typo in the algorithm 1, FibonacciTransfer. There should be space between Fibonacci and Transfer? If it is a standard term then it is OK.

Reply (v): Sir, the algorithm name is written as FibonacciTransfer intentionally, as it is an user defined function.

Review (vi) What if experiments are conducted on a variable size blocks. In this paper non overlapping blocks of sizes 8x8 are considered. What if the block size is 4x4 or greater than 8x8?

Reply (vi): Sir, we have included a new section (Section 3.1), on “computing the image processing block size”. This section gives an outline for fixing the block size as 8×8 for the experiments. If we use 4×4 block size, the embedding rate will obviously increase on bits per pixel basis, but the recovery rate may get reduced. We have highlighted in Section 3.7, that the proposed system is able to successfully recover over 90% of the images from various categories of USC SIPI dataset, except that of the Texture category, where the recovery rate is around 67%. This implies that the recovery gets affected by the random nature of pixels in the block. I.e., our ensemble approach works by classifying the decrypted blocks into natural and random blocks. In texture images, the pixels are highly random in nature. This affected the recovery rate. Similarly, when we try with very small blocks, there will be less pixels in the blocks and hence, distinguishing between the original block and the random blocks would become difficult. This will lead to less recovery rate. When the recovery of the image gets affected, that inturn reflects the bit error rate of hidden data extraction.

Review (vii) Also images are taken of size 512x512. It would be interesting to consider images of small and large sizes as well.

Reply (vii): We thank the reviewer for raising this comment. The reviewer may kindly note that the images in the USC SIPI image data set consists of 4 categories of images like Aerials, Sequences, Textures and Miscellaneous. The size of all the images under these categories are not the same. The image sizes vary from 256×256 to 1024×1024. For consistency, we standardize the images to size 512×512 before the actual processing. 

We thank all the constructive comments from Reviewer 1. We addressed all the comments from Reviewer 1 in the revised manuscript.

Reviewer 2 Report

Some comments as:

1)      Please improve the abstract and introduction to point better to the scientific contribution and novelty of the proposal and insert already the abstract quantitative data that demonstrate the benefits of your approach.

2)      Motivation and contribution sections should be added in the introduction.

3)      Is the presented framework in Fig. 1 represents a data-hiding mechanism?

4)      What is the utilized PRNG mechanism for generating a pseudo-random matrix?

5)      What about key K1, its size, range, etc.?

6)      Is the bitwise XOR operation sufficient for constructing a robust cipher image?

7)      Please revise lines 79-81.

8)      The stated steps in subsection "2.1. Hiding the secret information" are not clear, please provide a pseudocode algorithm for this part.

9)      Fibonacci transform is utilized for scrambling the cover image. Is this technique efficient in hiding data?

10)  The extraction process is not logical.

11)  By considering the extraction process is efficient, there is no security for the presented data hiding mechanism due to the utilized key is used for encrypting the cover image, not for the secret data.

12)  The embedding payload capacity is not sufficient for embedding large data.

13) Image quality analysis must be applied on the cover object and the stego object (sent object to the receiver).

Author Response

This paper presents a new way of secret embedding and recovery for RDH in encrypted images. Most of the conventional RDH schemes in encrypted images alter the pixel intensities of the encrypted image while trying to hide the secret information. The prime intention of the encryption algorithm is to secure the cover image. Thus, to compensate for the same, every RDH in encrypted images tries to achieve an entropy which is nearly equal to the entropy of the encrypted image. Thus the proposed algorithm is an effort that doesn’t compromise the entropy of encrypted images. This has been made possible by carefully designing the RDH scheme with an image scrambling algorithm, “Fibonacci transform” for hiding the secret information. Also, the recovery module is modeled through an ensemble approach with machine learning models. This has been another key point of the proposed scheme, which helps in an efficient lossless recovery of cover images and hence the recovery of the embedded secret.   

Some recommendations to improve the paper:

Review 1: Please improve the abstract and introduction to point better to the scientific contribution and novelty of the proposal and insert already the abstract quantitative data that demonstrate the benefits of your approach.

Reply 1: We thank the reviewer for this valuable comment. As per the comment, the abstract and the introduction sections are modified to make it better from the point of scientific contribution and novelty of the proposed work. Quantitative data that demonstrate the benefits of the proposed RDH scheme is also included in the abstract.

Review 2: Motivation and contribution sections should be added in the introduction

Reply 2: Two new subsections (Subsection 1.1 and Subsection 1.2) are introduced in the revised manuscript to highlight motivation and the contribution of the work in the introduction section.

Review 3: Is the presented framework in Fig. 1 represents a data-hiding mechanism?

Reply 3:  The reviewer may kindly note that Fig. 1 does not detail the data hiding mechanism, but the framework in Fig.1 shows the complete outline of the proposed RDH scheme, that includes the sender and receiver parts.

Review 4: What is the utilized PRNG mechanism for generating a pseudo-random matrix?

Reply 4: We have used the “rng” function to specify the seed and to control the random number generator used by “randi” function. This helps to generate the pseudo random matrix, which is of the same size as that of the cover image and whose values range in [0-255].  This matrix we have used for encryption purpose. The reviewer may kindly note that the authors are proposing a new RDH scheme in encrypted images. The proposed scheme will work for any encrypted images that are obtained through any encryption scheme. The bit-wsie XOR operation of an image with a pseudo-random matrix of same size is one way of generating the encrypted images and we have used this approach.

Review 5: What about key K1, its size, range, etc.?

Reply 5: The proposed RDH scheme is an RDH in encrypted images. Hence, the work mainly concentrates on the data hiding mechanism and the recovery part, with complete reversibility. Since the input to the system is an encrypted image, we can use any encryption algorithm to meet the purpose, which is not the goal of the proposed work.  

Thus, we have used a pseudo-random matrix, XORed with the cover image to generate the encrypted cover image. The whole RDH scheme works over this encrypted image. The Key K1 denotes the seed value used to generate this unique pseudo-random matrix. This key is also shared with the receiver to generate the same pseudo-random matrix, without which the receiver cannot recover the original cover image. The reviewer may note that proposed RDH scheme is not dependent on the encryption scheme.

Review 6: Is the bitwise XOR operation sufficient for constructing a robust cipher image?

Reply 6: We thank the reviewer for raising this question. We would like to convey that the goal of the proposed work is to hide the secret data inside an existing encrypted image and perform a complete reversibility along with the recovery of the hidden secret. Constructing a robust cipher image is altogether another problem that we can work on in the future. In this proposed work, we focus on the later part of the process that comes after receiving an encrypted image. That’s the reason for using bitwise XOR operation to generate an encrypted image. It should be noted that we could use any encryption algorithm to encrypt the image with a condition that the key should be shared with the receiver to decrypt it.

Review 7: Please revise lines 79-81.

Reply 7: We have revised the lines 79-81 and the updated contents are at lines 137-140 in the latest version. 

Review 8: The stated steps in subsection "2.1. Hiding the secret information" are not clear, please provide a pseudocode algorithm for this part.

Reply 8: As per the comments, we have included a pseudocode for the fibonacci transfer function as Algorithm 2. Algorithms 1 and 2 jointly perform the hiding of the secret information. 

Review 9: Fibonacci transform is utilized for scrambling the cover image. Is this technique efficient in hiding data

Reply 9: Every RDH scheme has some mechanism to hide the data. In the proposed work, there are two purposes for using the Fibonacci transform algorithm:

  1. To hide the data in the encrypted image.
  2. To retain the pixel intensities of the encrypted image even after embedding the secret data. This is one of the key contributions of the proposed work.

The only issue of using Fibonacci transform for hiding data is in having a fixed payload. This depends on the periodicity property of the Fibonacci transform. It is still efficient to use the Fibonacci transform to embed the secret data as it is very difficult to identify any changes in the data hidden image and the encrypted image.

Review 10: The extraction process is not logical.

Reply 10: The extraction of the hidden secret data follows recovery of the original block. The sender is transferring a data hidden encrypted image to the receiver. Receiver processes the received image block by block. The original block is initially recovered from the data hidden encrypted block by checking the voting from the ensemble models on the decrypted blocks. Once the original block is identified, the receiver has two blocks: 1. Original block and 2. Data hidden encrypted block. Since both the blocks are present with the receiver, he can use the Fibonacci transform to find the hidden secret data. 

Review 11: By considering the extraction process is efficient, there is no security for the presented data hiding mechanism due to the utilized key is used for encrypting the cover image, not for the secret data.

Reply 11: We thank the reviewer for raising this comment. It is true that that the key is used only for encrypting the image and there is no other key involved to secure the secret data. But in this RDH approach, we are not altering any of the pixel intensities of the encrypted image for hiding the data. That’s why we didn’t use any encoding scheme or any compression algorithms to secure the data. We assume that the communication channel is robust and free from noises.  

Review 12: The embedding payload capacity is not sufficient for embedding large data.

Reply 12: Sir, it is true that the payload of the proposed work may not be sufficient to embed a very large data stream. But the proposed work is an effort to come up with a new approach where the encrypted pixels are not modified while embedding the data. So that in practical applications, the efficiency of the encryption algorithm is not compromised for hiding the data. Moreover, we were able to obtain a payload of 0.0625 bits per pixel on a 512×512 cover image, which surpassed few of the recent state of the art similar RDH schemes. 

Review 13: Image quality analysis must be applied on the cover object and the stego object (sent object to the receiver).

Reply 13: The reviewer may kindly note that the image quality analysis like PSNR and SSIM on cover object and the stego object is significant in the RDH schemes, where the data is embedded on the natural image prior to the encryption stage. Our proposed scheme is an RDH in encrypted image and hence there is no stego object. Thus we can’t perform image quality analysis on this basis.

We thank all the constructive comments from Reviewer 2. We addressed all the comments from Reviewer 2 in the revised manuscript.

Reviewer 3 Report

In the introduction section, the author explains the difference between RDH and watermarking, especially fragile watermarking. I think this explanation is self-explanatory, perhaps it needs to be shortened. My suggestion is that the author also discusses the difference between steganography so that readers can more clearly grasp the purpose of RDH.

In the Introduction the author has explained several standard RDH methods, their weaknesses and strengths. But the author failed to convey the problem what happened so that he had to propose the RDH method on encrypted and Fibonaci images. What are the advantages of why encryption must be done first? And what are the advantages of the Fibonaci method?

The proposed RDH technique performs embedding on the encrypted image, what is its purpose and advantages? What is the difference with other RDH methods not using encryption techniques? For example in the following papers:

*An Improved Histogram-Shifting-Imitated reversible data hiding based on HVS characteristics

*High fidelity based reversible data hiding using modified LSB matching and pixel difference

*High capacity reversible and secured data hiding in images using interpolation and difference expansion technique

*High-fidelity reversible data hiding by Quadtree-based pixel value ordering

In data hiding generally consists of several main criteria, namely imperceptibility, payload, security, and sometimes there are some who are also concerned with robustness. Why is the main goal only on the payload? Even though you combine it with encryption techniques.

The encryption process is actually quite simple, only using a substitution process with an XOR operation based on a key and a pseudo random generator. What is the main purpose of this encryption? Is it to increase security or other effects?

You haven't mentioned a scientific reason, why should the 8×8 block be chosen?

By embedding the encrypted image and if the ecryted image after embedding cannot be transformed to a form identical to the original image, the imperceptibility quality cannot be calculated, unless you calculate the imperceptibility quality based on PSNR and SSIM encrypted image after and before embedding. Section 3.3 is more appropriately called a restored cover image analysis.

What is the difference in the discussion of the results in sections 3.3 and 3.6? Why did the sample restored cover image on 3.3 work completely, while on 3.6 it didn't work at all, what affects this value?

In sections 3.4 and 3.5, measurements related to image encryption are carried out. Why is this necessary in your proposed RDH? I think this has something to do with encryption security, why only the entropy and histogram are measured? Why not measure other measuring instruments such as differential analysis, etc.?

Why is the comparative study based on payload only? can other criteria in other hiding data not be met or compared? If you can't compare why? is it because the proposed method is considered a new technique in RDH?

Writing table names is not complete, for example Table 2, is this PSNR and SSIM on what images? In Table 4, what is measured? Need to be written in the table title.

Overall presentation of the proposed method still needs to be clarified again, what the problem is, what is the purpose and what is the main contribution of the proposed method. This is because there are so many developments and goals from RDH so that readers need to know the direction and purpose of the proposed method. Measurements also need to be adjusted to the direction and objectives of the RDH.

Author Response

Review 1: In the introduction section, the author explains the difference between RDH and watermarking, especially fragile watermarking. Ithink this explanation is self-explanatory, perhaps it needs to be shortened. My suggestion is that the author also discusses the difference between steganography so that readers can more clearly grasp the purpose of RDH.

Reply 1: We thank the reviewer for suggesting to include steganography that will help readers to have a better understanding of the importance of RDH. As per the suggestion, a short note on the steganography is added in the introduction and we have also reduced the contents of watermarking a little (Please refer line nos. 45 to 52 in the revised version of the manuscript). 

Review 2: In the Introduction the author has explained several standard RDHmethods, their weaknesses and strengths. But the author failed to convey the problem what happened so that he had to propose theRDH method on encrypted and Fibonacci images. What are the advantages of why encryption must be done first? And what are the advantages of the Fibonacci method?

Reply 2: By considering the valuable suggestion from the reviewer, we have modified the contents of the introduction to convey the problems in the existing literature and justify the needfulness of the proposed work. Advantage of using encrypted image as a cover image and that of Fibonacci transform is also covered.  (Please refer line nos. 71 to 123 in the received manuscript).

Review 3:  The proposed RDH technique performs embedding on the encrypted image, what is its purpose and advantages? What is the difference with other RDH methods not using encryption techniques? For example in the following papers:

  • An Improved Histogram-Shifting-Imitated reversible data hiding based on HVS characteristics 
  • High fidelity based reversible data hiding using modified LSBmatching and pixel difference 
  • High capacity reversible and secured data hiding in images using interpolation and difference expansion technique
  • High-fidelity reversible data hiding by Quadtree-based pixel value ordering

Reply 3: We thank the reviewer for raising this comment.  All the four papers mentioned in the Review 3 comment use the original image or natural image as the cover image. We have cited the papers in the introduction section (line no. 76 in the updated version).

The reviewer may kindly note that there are three main entities involved in any RDH scheme. They are the content owner, the data hider and the receiver. In some scenarios, we may have to maintain confidentiality between the content owner and the data hider. In such cases, the RDH in encrypted images will be very useful. For example, In a medical scenario, we can think the content owner to be a doctor and the data hider to be a clerk. Say, the doctor doesn’t wish to disclose the patient’s image to the clerk. Here, the doctor will encrypt the image and handover the encrypted file to the clerk. The data hiding process will be carried out by the clerk before communicating it to the expert doctor or the receiver. 

Review 4: In data hiding generally consists of several main criteria, namely imperceptibility, payload, security, and sometimes there are some who are also concerned with robustness. Why is the main goal only on the payload? Even though you combine it with encryption techniques.

Reply 4: It is true that data hiding techniques may concentrate on criteria like imperceptibility, payload, security and sometimes robustness. But it is very rare that a single RDH scheme would address all the criteria. The problem which we try to address here is by assuming that the channel is robust and noise free. Because, addressing robustness would become another research problem in itself. Also, criteria like imperceptibility is not applicable in the proposed scheme. Mostly, researchers in the field addressing data hiding problems would highlight imperceptibility when they embed the data over the original image and then go for encryption. But in our case, the input itself is an encrypted image. 

In the proposed approach, though we have highlighted a comparison concerning the payload, every scheme in the comparison table is also analyzed on the quality of the recovered cover image as well. All the schemes are completely reversible, hence their PSNR = ∞ and SSIM = 1. That’s why they have not been highlighted. 

Payload was not the only goal of the proposed work, but we were also trying to support the security of the encrypted image through the scrambling algorithm “Fibonacci transform”. Ofcourse, the complete recoverability can’t be accomplished if we were not able to maintain a good tradeoff between the payload and the quality of the recovered image. 

Review 5: The encryption process is actually quite simple, only using a substitution process with an XOR operation based on a key and a pseudo random generator. What is the main purpose of this encryption? Is it to increase security or other effects?

Reply 5: We thank the reviewer for raising this question. We would like to convey that the proposed scheme is an RDH in encrypted image. So, the input to the data hider will be an encrypted image and the problem which we try to address is data hiding and recovery on this encrypted image. Since, we are not majorly concerned with the encryption algorithm used, we tried using simple XOR operation with pseudo random matrix to generate the encrypted image. This has nothing to do with the security or any other effects pertaining to the proposed system.

Review 6: You haven't mentioned a scientific reason, why should the 8×8 block be chosen?

Reply 6: We thank the reviewer for raising this question. The proposed scheme works by processing the encrypted image into non overlapping sub blocks of the same size. The payload of the scheme depends on the amount of data that can be embedded in each block. In the proposed scheme, data hiding is achieved via the Fibonacci transform and the embedding rate depends on the periodicity of the Fibonacci transform. Based on the transform matrix adopted in the fibonacci transform function, the periodicity is as follows:

Sl. No.

Block Size

Periodicity

Bits embedded in a block

Payload achieved (bpp)

1

8 × 8

16

4

0.0625

2

16 × 16

32

5

0.0195

3

32 × 32

64

6

0.0059

Based on the periodicity we understand that the maximum payload achievable is 0.0625 bpp. Also, by taking the recoverability factor into account, we have adopted 8 × 8 to be the block size for processing the images. We once again thank the reviewer for raising this comment. Accordingly, we have added a new subsection under the experimental study with the title, “Computing the image processing block size”. The newly added subsection number is 3.1.

Review 7:  By embedding the encrypted image and if the encrypted image after embedding cannot be transformed to a form identical to the original image, the imperceptibility quality cannot be calculated, unless you calculate the imperceptibility quality based on PSNR and SSIM encrypted image after and before embedding. Section 3.3 is more appropriately called a restored cover image analysis.

Reply 7: We thank the reviewer for this comment. Taking into account, we have changed the title of section 3.3 to restored cover image analysis (the section is now 3.4 in the updated version).

Review 8: What is the difference in the discussion of the results in sections 3.3 and 3.6? Why did the sample restored cover image on 3.3 work completely, while on 3.6 it didn't work at all, what affects this value?

Reply 8: Sir, In the whole paper we have tried to highlight the values obtained on four specific images called Airplane, Baboon, Boat and Peppers. Most of the researchers in this field used to quote their results concerning these images in their work due to their varying properties. Hence, to make it competitive, we have used the same images to highlight our results as well. 

Section 3.3 only shows the quality of the recovery of these 4 images. The PSNR and SSIM values in Table 2 under section 3.3 indicate that all the 4 images are recovered successfully without any loss. But section 3.6 shows the percentage of recovered images on the whole USC-SIPI image dataset. The USC-SIPI database consists of 210 images and they fall under four categories: Aerials, Sequences, Textures and Miscellaneous. We have analyzed the recovery of all images under these categories and disclosed the results in section 3.6. 

As we can observe in section 3.6, over 90% of the images under Aerials, Sequences, and Miscellaneous categories were successfully recovered, whereas the recovery rate is around 67% on Textures. As we know our ensemble model tries to identify the original block from the decrypted blocks. In other words, the models classify the decrypted blocks into either original or encrypted blocks. The pixels in a texture image are highly random in nature. Hence, it is not able to classify the decrypted blocks properly. This has been the main cause of reduction in the recovery rate of Textured images (the section 3.3 is section 3.4 and section 3.6 is section 3.7 in the updated version of the manuscript). 

Review 9: In sections 3.4 and 3.5, measurements related to image encryption are carried out. Why is this necessary in your proposed RDH? I think this has something to do with encryption security, why only the entropy and histogram are measured? Why not measure other measuring instruments such as differential analysis, etc.?

Reply 9: We thank the reviewer for raising this comment. It is true that sections 3.4 and 3.5 have been discussed to show the retainment of the encryption security after the data hiding process. This section claims that the proposed reversible data hiding scheme not compromising the the encryption efficiency parameters. Entropy analysis and histogram analysis are the two well-known efficiency parameters that can be used to evaluate the randomness of the encrypted image. We have not considered differential analysis since we are using a stream cipher based image encryption process. The reviewer may kindly note that our interest is not to design or improve the image encryption process. Since we used a basic image encryption scheme to generate the encrypted image, and further, it will be considered as the input for proposed reversible data hiding scheme. Our RDH scheme is compatible with any encryption process. 

 Note: The reviewer may kindly note that in the revised manuscript, the section 3.4 is moved to the section 3.5 and the section 3.5 is moved to the section 3.6 due to the changes incorporated at the beginning of the section 3.

Review 10: Why is the comparative study based on payload only? can other criteria in other hiding data not be met or compared? If you can't compare why? is it because the proposed method is considered a new technique in RDH?

Reply 10: The reviewer may kindly note that all the schemes in the comparative study table are also compared on the quality recovery parameters like PSNR and SSIM also. Every scheme compared is able to recover the original cover image without any loss implying the PSNR to be ∞ and SSIM to be 1. Since they differ only in their payload, we have highlighted them in a separate table.

Review 11: Writing table names is not complete, for example Table 2, is thisPSNR and SSIM on what images? In Table 4, what is measured?Need to be written in the table title.

Reply 11: We thank the reviewer for giving valuable comments on the Table name. This will improve the readability. Accordingly, we have modified the table names of Table 1,  Table 2, Table 3 and  Table 4 (Table 1,  Table 2, Table 3 and  Table 4 are modified to Table 2, Table 3, Table 4 and Table 5 respectively in the updated version) .

Review 12: Overall presentation of the proposed method still needs to be clarified again, what the problem is, what is the purpose and what is the main contribution of the proposed method. This is because there are so many developments and goals from RDH so that readers need to know the direction and purpose of the proposed method. Measurements also need to be adjusted to the direction and objectives of the RDH.

Reply 12: We thank the reviewer for giving an insight on the overall presentation. To better the presentation, we have added novelty to the Abstract, updated Introduction section by including steganography to better understand the importance of RDH, added disadvantages of using the original image as such as a cover image, advantage of using encrypted image as a cover image, advantage of using Fibonacci transform for hiding data, motivation and contributions of the work are also included. As per the reviewers comments, we have modified the manuscript and we hope that now the proposed work has improved its overall presentation.

We thank all the constructive comments from Reviewer 3. We addressed all the comments from Reviewer 3 in the revised manuscript.

Reviewer 4 Report

rephrase the equation appropriate in mathematical notation 

Author Response

Rephrase the equation appropriate in mathematical notation.

Reply 1: We thank the reviewer for the positive feedback on the manuscript. By considering the valuable suggestions from the reviewer, we have revised the equations in the manuscript.

Reviewer 5 Report

A well written paper with a clear methodology, however some issues are detected:

Out of 12 SoA RDH schemes which are presented only 6 are placed within the last 5 years. A more recent selection would be strongly recommended.

Out of the 6 most recent ones [25-30], 3 schemes are from the same First Author as this paper. I d strongly recommend including cited journal papers from other authors to enhance validity of results.

Comparison with relevant work in the field should be more elaborated, describing also the motivation and scope of this work and how this addresses an identified shortcoming in relevant work where a significant amount of literature exist

Author Response

A well written paper with a clear methodology, however some issues are detected:

Review 1. Out of 12 SoA RDH schemes which are presented only 6 are placed within the last 5 years. A more recent selection would be strongly recommended.

Reply 1: We thank the reviewer for the valuable suggestion. By considering this comment, we discussed a few more recent works (published after 2020) in the literature and in addition, we have also included a comparison of 2 more recent schemes from other authors in the revised manuscript.

Review 2. Out of the 6 most recent ones [25-30], 3 schemes are from the same First Author as this paper. I d strongly recommend including cited journal papers from other authors to enhance validity of results.

Reply 2: We thank the reviewer for posting this valid comment. Accordingly, we have extended the comparative study by including a comparison of two more recent schemes from other authors to enhance the validity of results.

Review 3. Comparison with relevant work in the field should be more elaborated, describing also the motivation and scope of this work and how this addresses an identified shortcoming in relevant work where a significant amount of literature exist.

Reply 3: We thank the reviewer for the positive feedback on the manuscript. By considering the valuable comment from the reviewer, we have elaborated the experimental study section. The motivation and scope of this work are more precisely brought into the introduction section of the revised manuscript.

We thank all the constructive comments from Reviewer 5. We addressed all the comments from Reviewer 5 in the revised manuscript.

Round 2

Reviewer 3 Report

Responding to the author's answer from points 1 and 2, I see that the revised results are a little better. But the explanation is still too general. The author cannot explain why RDH-EI can be safer. The meaning of being safer here needs to be clearer and clearer. More secure in terms of detectability or decryption? Is it safer for covers or for embedded messages? The author should be more clear in explaining whether cover encryption is also used to protect the cover. If the reason for encryption is for cover security, why is the encryption method simple and only measured by entropy and histograms? If the reason is to increase undetectability, what is the logical reason that encrypted images are more difficult detected by steganalysis? As a note, undetectability is measured by steganalysis, while PSNR/SSIM is to measure imperceptibility. Then, is there a relationship between the security improvement with the use of encrypted images as covers?

The author's answer in point 2 states that the goal is "designing better RDH schemes, that makes a good trade-off between the payload and the quality of the recovered cover image", but here the author does not explain the problem of recovered cover image quality in the previous method. What is the problem? And when compared with references [22,28,29], the payload of the proposed method is still much smaller. Why are references [22,28,29] not also used as a comparison, but instead include other references that have not been discussed before?

It should be noted that data hiding/RDH/steganography methods are currently developing rapidly, and have advantages in only one or a several criteria. The author must explain the urgency of why to design the RDH. My suggestion is that the author must be able to explain the problem and the importance of having to design a good trade-off between payload and recovered cover image quality. My suggestion is that the author should explain in more detail in a special section, namely related work.

Responding to point 3, I don't ask the author to have to cite the paper if he can't explain it in detail. I want the author to be able to explain in more detail and scientifically so that readers can understand the purpose of the method you are proposing. With this explanation, the Author can explain the reasons why he chose to design an RDH with a good trade-off between payload and recovered cover image quality, not other criteria such as imperceptibility, security/undetectability or something else?

In response to point 4, I agree that it is nearly impossible that all criteria improve. What I mean is I want you to explain why you have the criteria you mentioned above. Furthermore, regarding the level of cover reversibility with PSNR = ∞ and SSIM = 1, this value is reasonable and should be achieved by various RDH methods so that it can be called RDH, so there is no significant contribution to this.

Still responding to point 4, "Payload was not the only goal of the proposed work, but we were also trying to support the security of the encrypted image through the scrambling algorithm "Fibonacci transform", with this statement the security must be proven and compared with other encryption techniques.

In the manuscript, what security is meant is "The use of Fibonacci transform algorithm for data hiding helps to retain the pixel intensities of the encrypted image. Thus preserving the same entropy and histogram of the encrypted image and the image after hiding the data. This helps to support the encryption efficiency and security established via the encryption algorithm."? Is that the same as minimizing distortion after embedding? Then what's the difference with imperceptibility? Why is it measured by a histogram and entropy? The goal is to increase the security of the message embedding or cover encryption security? If it's to improve the security of embedding messages, why not test it with steganalysis?

Responding to point 5, I conclude that the Author stated that there was a problem recovering the cover after it was encrypted. This is not explained in detail in the introduction. The author should explain some related research in detail and state that they have this problem. In addition, the encryption method has various characteristics, of course, will affect the recovery technique. I do not agree that the encryption technique was chosen haphazardly, so the author needs to provide more scientific reasons.

Authors shouldn't just state that embedding encrypted images is safer. Statements on lines 77-81 need to be added with a citation. If lines 77-81 are pure statements from the author, I do not agree with this, many steganographic methods with natural images and payloads below 1BPP are proven to be safe from various steganalytic attacks. What do "third party attacks" mean?

Responding to point 6, if the 8x8 block has a greater payload  than 16x16 or 32x32. Why not choose 4x4 or 2x2 blocks? By reading the table, logically 4x4 or 2x2 will produce a bigger payload.

Responding to point 8, this proves that recovering 100% cover is a problem in RDH-EI, as you alluded to in section 1. Unfortunately this explanation is not as detailed and explicit as in previous research (especially refs [22,28,29] ) ran into the same problem. If this is a problem, why is it called RDH, even though the RDH method should be able to recover 100%. Then if in previous research this was highlighted, why is there no comparison with previous research in this section?

In response to point 9, unfortunately you have not explained the importance of measuring efficiency in sections 3.4 and 3.5. This needs to be explained and given a citation.

Responding to point 10, I agree that recovered has been measured, but this has not been compared to previous research. The payload trade-off contribution and recovered cover image quality cannot be proven if only the payload is compared to previous research. And, if the previous research also got 100% recovered, it means that this is not an RDH problem, it is actually a RDH requirement. And when reading references [22,28,29], the method proposed in the research was also successfully 100% recovered and had a payload > 2BPP.

Responding to point 12, I see that the results of the revision are still not satisfactory. This is evidenced by the many comments and questions that I have written above. I suggest that the author can prove and explain that the urgency of the research and the contribution of the proposed method are quite significant and feasible. The presentation flow needs to be improved as a whole so as not to raise a lot of questions.

Additional question, why is Table 5 compared to references [22,28,29]? Why was reference [34-46] not mentioned before, and suddenly appears in section 3.8? Need to explain why compared with these studies?

Author Response

REPLY TO REVIEWER 3

Responding to the author's answer from points 1 and 2, I see that the revised results are a little better. But the explanation is still too general. The author cannot explain why RDH-EI can be safer. The meaning of being safer here needs to be clearer and clearer. More secure in terms of detectability or decryption?

Is it safer for covers or for embedded messages? 

The author should be more clear in explaining whether cover encryption is also used to protect the cover. 

If the reason for encryption is for cover security, why is the encryption method simple and only measured by entropy and histograms? 

If the reason is to increase undetectability, what is the logical reason that encrypted images are more difficult detected by steganalysis? As a note, undetectability is measured by steganalysis, while PSNR/SSIM is to measure imperceptibility. Then, is there a relationship between the security improvement with the use of encrypted images as covers?

The reviewer may kindly note that the detection of the presence of hidden secret information becomes difficult, when the data is embedded in an encrypted image. Hence, it is safer to use RDH-EI for securing the secret data. 

Cover image encryption does secure the cover image. That is to say, the random nature of the encrypted cover image pixels would not reveal any information [provide confidentiality]  about the original cover image. Since the hidden message is embedded into the image after encryption, the message is also secure. These points are added in the revised manuscript.

RDH-EI approach helps in a secure transmission of the information. Altogether the whole system would make it safer for both the cover image and the secret data. There are different RDH methods of which RDH-EI is one of its kind. We have proposed a RDH over an encrypted image which doesn’t hinder the security of the encryption process. The reviewer may kindly note that we are not assessing the security of the encryption algorithm. The entropy and histogram analysis is done to convey that the pixel distribution in the encrypted image is retained even after hiding the secret data. Hence, the security that is imposed on the cover image through any encryption process doesn’t get altered. The reviewer may kindly note that the proposed RDH scheme is compatible with any encryption process. 

The RDH schemes are basically three categories: RDH in natural images, RDH in encrypted images and RDH through encryption. In this manuscript, we propose a new RDH scheme in encrypted image. The RDH in encrypted images are suitable when we need to ensure the confidentiality of the cover image from the data hider [those who embed secret data]. Most of the existing RDH schemes in encrypted images are lagging in terms of embedding rate. By considering this as a challenge, we introduced a novel RDH scheme for encrypted images. On top of this, steganalysis is normally performed on cover images when it is in natural form. So the steganalysis is difficult on the proposed framework. These points are added in the revised manuscript. 

The author's answer in point 2 states that the goal is "designing better RDH schemes, that makes a good trade-off between the payload and the quality of the recovered cover image", but here the author does not explain the problem of recovered cover image quality in the previous method. What is the problem? And when compared with references [22,28,29], the payload of the proposed method is still much smaller. Why are references [22,28,29] not also used as a comparison, but instead include other references that have not been discussed before?

As the reviewers mentioned, the embedding rate provided by the schemes in [22, 28, 29] are better than the proposed scheme. But the key difference between the proposed scheme and  the schemes reported in [22, 28, 29] is that the proposed scheme retains the pixel distribution. This work mainly focussed to introduce a RDH scheme in encrypted images which is capable to retain the pixel distribution of the encrypted image with best possible embedding rate. By considering the valuable comments from the reviewer, we have commented on the results of the schemes mentioned here in the revised manuscript [Section: 4.8, Line nos. 401-405]. 

It should be noted that data hiding/RDH/steganography methods are currently developing rapidly, and have advantages in only one or a several criteria. The author must explain the urgency of why to design the RDH. My suggestion is that the author must be able to explain the problem and the importance of having to design a good trade-off between payload and recovered cover image quality. My suggestion is that the author should explain in more detail in a special section, namely related work.

We thank the reviewer for this valuable point to improve the readability of the manuscript. By considering this review, we split the introduction part into two and the second part solely discusses the related works in this domain. We highlighted the need for designing a good RDH scheme in this section and its importance in the Introduction and Related work sections. 

Responding to point 3, I don't ask the author to have to cite the paper if he can't explain it in detail. I want the author to be able to explain in more detail and scientifically so that readers can understand the purpose of the method you are proposing. With this explanation, the Author can explain the reasons why he chose to design an RDH with a good trade-off between payload and recovered cover image quality, not other criteria such as imperceptibility, security/undetectability or something else?

In the revised manuscript, we have discussed the details pertaining to the needfulness of the proposed work in the Introduction section. Hope this will improve the readability of the manuscript.  

In response to point 4, I agree that it is nearly impossible that all criteria improve. What I mean is I want you to explain why you have the criteria you mentioned above. Furthermore, regarding the level of cover reversibility with PSNR = ∞ and SSIM = 1, this value is reasonable and should be achieved by various RDH methods so that it can be called RDH, so there is no significant contribution to this.

We thank the reviewer for raising this comment. It is true that in a data hiding scheme to be called as a RDH, the cover image should be completely recovered. Hence the PSNR = ∞ and SSIM = 1. We have tried to maintain the pixel distribution of the encrypted image while embedding the data through our proposed scheme and managed to achieve an acceptable payload without compromising the recoverability.

Still responding to point 4, "Payload was not the only goal of the proposed work, but we were also trying to support the security of the encrypted image through the scrambling algorithm "Fibonacci transform", with this statement the security must be proven and compared with other encryption techniques.

We kindly request the reviewer to note that we have not designed any encryption algorithm in the proposed scheme. In the proposed work, we have focused on hiding the message in a given encrypted image and recover both the cover image and the hidden data efficiently. We were pointing out that whatever encryption algorithm a user employs, the proposed system of RDH will not compromise its efficiency.

In the manuscript, what security is meant is "The use of Fibonacci transform algorithm for data hiding helps to retain the pixel intensities of the encrypted image. Thus preserving the same entropy and histogram of the encrypted image and the image after hiding the data. This helps to support the encryption efficiency and security established via the encryption algorithm."? Is that the same as minimizing distortion after embedding? Then what's the difference with imperceptibility? Why is it measured by a histogram and entropy? The goal is to increase the security of the message embedding or cover encryption security? If it's to improve the security of embedding messages, why not test it with steganalysis?

As far as we know, the imperceptibility analysis is possible when we embed the data on a natural image. Since we perform data hiding in encrypted images, we have used entropy and histogram analysis. Further, entropy analysis and histogram analysis highlight that the proposed scheme retains the same pixel distribution even after data hiding.

The reviewer may kindly note that the scrambling techniques like Rubik's cube algorithm, R-Prime Shuffle, Arnold’s cat map, Fibonacci transform, etc. itself are used to provide security for the images through predefined scrambling process. So the scrambling process that we perform on an encrypted image will surely strengthen the system. 

Responding to point 5, I conclude that the Author stated that there was a problem recovering the cover after it was encrypted. This is not explained in detail in the introduction. The author should explain some related research in detail and state that they have this problem. In addition, the encryption method has various characteristics, of course, will affect the recovery technique. I do not agree that the encryption technique was chosen haphazardly, so the author needs to provide more scientific reasons.

We would like the reviewer to kindly note that the simple encryption technique through XOR operation certainly helps in recovering the original cover at the receiver end. But we could also use any other encryption techniques to encrypt the cover image before data hiding. In this case the decryption key corresponding to the encryption algorithm should be available with the receiver to decrypt the cover image correctly.

Authors shouldn't just state that embedding encrypted images is safer. Statements on lines 77-81 need to be added with a citation. If lines 77-81 are pure statements from the author, I do not agree with this, many steganographic methods with natural images and payloads below 1BPP are proven to be safe from various steganalytic attacks. What do "third party attacks" mean?

By considering the valuable comment from the reviewer, we removed the statement where we claimed that “embedding encrypted images are safer” and we thank the reviewer for correcting the same with additional information. The term “Third party attack”, we used, indicates the attackers [unauthorized people who are trying to extract the hidden information]. We have modified the term  “third party attack” to unauthorized access to give more clarity for the readers. We have also added citations to the points mentioned in the statements 77-81 [Line nos. 60-64 in the revised manuscript].

Responding to point 6, if the 8x8 block has a greater payload  than 16x16 or 32x32. Why not choose 4x4 or 2x2 blocks? By reading the table, logically 4x4 or 2x2 will produce a bigger payload.

The reviewer may kindly note that, though the payload while processing the image with smaller block size will yield a better payload, the reversibility criteria should also be good. In our scheme, we can embed 3 bits of information in a 4×4 block, which gives an improved payload of 0.1875 bpp. But when we computed the recovery rate, only 10 images from 64 texture images were successfully recovered, which is only 15.625%. In this case, the overall recovery rate on the USC SIPI image data set is only 52.86%. Which is very less compared to the 88.1% of recovery rate produced while processing with 8×8 block size. Hence we confined our experimental study to blocks with size of 8x8 pixels.

Responding to point 8, this proves that recovering 100% cover is a problem in RDH-EI, as you alluded to in section 1. Unfortunately this explanation is not as detailed and explicit as in previous research (especially refs [22,28,29] ) ran into the same problem. If this is a problem, why is it called RDH, even though the RDH method should be able to recover 100%. Then if in previous research this was highlighted, why is there no comparison with previous research in this section?

100% recoverability is a requirement for any RDH scheme, but, a number of RDH schemes are there which can recover the images with high probability but not always [the scheme like [23] Zhang, X. Reversible data hiding in encrypted image. IEEE signal processing letters. 2011]. The reviewer may kindly note that our main objective is to introduce a new RDH scheme which will not change the pixel distribution [such schemes are very rare [44, 50] in the literature]. In this we improved those schemes by introducing an ensemble learning approach. We have compared the proposed scheme with existing schemes in Table 5 and note that in Table 5, only the schemes [44, 50] can only retain the pixel distribution. 

In response to point 9, unfortunately you have not explained the importance of measuring efficiency in sections 3.4 and 3.5. This needs to be explained and given a citation. 

We thank the reviewer for pointing out the need to specify the importance of measuring parameters in sections 3.4 and 3.5. We have incorporated the required changes in the sections as mentioned by adding citations. The sections are 4.4 and 4.5 in the revised manuscript.

Responding to point 10, I agree that recovered has been measured, but this has not been compared to previous research. The payload trade-off contribution and recovered cover image quality cannot be proven if only the payload is compared to previous research. And, if the previous research also got 100% recovered, it means that this is not an RDH problem, it is actually a RDH requirement. And when reading references [22,28,29], the method proposed in the research was also successfully 100% recovered and had a payload > 2BPP.

We kindly agree to the point mentioned by the reviewer. The embedding rate from the schemes [22,28,29] are greater than the embedding rate from the proposed scheme. But the reviewer may kindly note that the highlight of the proposed scheme is that we have retained the pixel distribution of the cover image even after the data is hidden. This point is added in the revised manuscript. [References 22,28,29 are 13,14,15 in the revised manuscript]

Responding to point 12, I see that the results of the revision are still not satisfactory. This is evidenced by the many comments and questions that I have written above. I suggest that the author can prove and explain that the urgency of the research and the contribution of the proposed method are quite significant and feasible. The presentation flow needs to be improved as a whole so as not to raise a lot of questions.

We thank the reviewer for the detailed comment which certainly helps us to improve the quality of the manuscript. We have provided the information for the need of the proposed work in the Introduction section [line nos. 80-93]. We have completely modified the Introduction section by including Related work as a separate section. We tried to improve the flow of the whole manuscript by articulating the discussions in a better way. 

Additional question, why is Table 5 compared to references [22,28,29]? Why was reference [34-46] not mentioned before, and suddenly appears in section 3.8? Need to explain why compared with these studies?

We thank the reviewer for commenting on Table 5. All the schemes that were presented in Table 5 are RDH schemes on encrypted images. A fair comparison with all the schemes can’t be made as the proposed scheme intends to retain the pixel intensities of the encrypted image while data embedding. However, we have discussed [22,28,29] in the revised manuscript [note that, in revised manuscript references [13, 14, 15]] schemes [line nos. 152-159]. The schemes [34-46] that we have considered for comparison are also introduced in the Introduction and Related work section.

Reviewer 5 Report

I d like to thank the authors for providing a significantly improved version for the article. No further issues from my side.

Author Response

I d like to thank the authors for providing a significantly improved version for the article. No further issues from my side.

We thank the reviewer for the positive comment on the revised manuscript.

Round 3

Reviewer 3 Report

In this second revision, I see that the presentation of this manuscript is much better. But there is still one thing that needs to be clarified.

In the following sentence "Basically, the goal of steganography lies in the secure communication of the secret data and not the cover. This also implies that we could adopt any cover to transmit the secret data. The receiver will extract the hidden message from the cover and during the process, the cover may get distorted which is not a concern from a steganographic point of view. This makes reversible data hiding (RDH) different from steganography".

I agree and it's logical, indeed the embedding process will produce distortion, in encrypted images the distortion will be difficult to detect by human vision. But this is not necessarily the case with steganalysis. It would be very good if this is also proven. Even if it's not directly proven, maybe you can add one or two pieces of literature that have tried to prove it.

Author Response

In the following sentence "Basically, the goal of steganography lies in the secure communication of the secret data and not the cover. This also implies that we could adopt any cover to transmit the secret data. The receiver will extract the hidden message from the cover and during the process, the cover may get distorted which is not a concern from a steganographic point of view. This makes reversible data hiding (RDH) different from steganography".

I agree and it's logical, indeed the embedding process will produce distortion, in encrypted images the distortion will be difficult to detect by human vision. But this is not necessarily the case with steganalysis. It would be very good if this is also proven. Even if it's not directly proven, maybe you can add one or two pieces of literature that have tried to prove it.

Steganography and steganalysis are two sides of the same coin. Steganalysis for hidden message extraction is quite difficult when the data hider embeds less information and/or when the image is in the encrypted form. Machine learning-based approaches are explored recently for steganalysis to overcome these challenges. We updated the manuscript by including proper references to support these points. 

The reviewer may kindly note that in the revised manuscript, the authors cited a few more additional reference to introduce the data hiding schemes which focuses to reduce the quality degradation of the cover image for better security. 

We thank the reviewer for this valuable comment which will make the manuscript technically strong.